# AdaPlanner: Adaptive Planning from Feedback with Language Models

**Haotian Sun**[1],[*]  **Yuchen Zhuang**[1],[*]  **Lingkai Kong**[1],  **Bo Dai**[1],  **Chao Zhang**[1]

[1] Georgia Institute of Technology

{haotian.sun, yczhuang, lkkong, chaozhang}@gatech.edu,  bodai@cc.gatech.edu

## Abstract

Large language models (LLMs) have recently demonstrated the potential in acting as autonomous agents for sequential decision-making tasks. However, most existing methods either take actions greedily without planning or rely on static plans that are not adaptable to environmental feedback. Consequently, the sequential decision-making performance of LLM agents degenerates with problem complexity and plan horizons increase. We propose a closed-loop approach, *AdaPlanner*, which allows the LLM agent to refine its self-generated plan adaptively in response to environmental feedback. In AdaPlanner, the LLM agent adaptively refines its plan from feedback with both *in-plan* and *out-of-plan* refinement strategies. To mitigate hallucination, we develop a code-style LLM prompt structure that facilitates plan generation across a variety of tasks, environments, and agent capabilities. Furthermore, we propose a skill discovery mechanism that leverages successful plans as few-shot exemplars, enabling the agent to plan and refine with fewer task demonstrations. Our experiments in the ALFWorld and MiniWoB++ environments demonstrate that AdaPlanner outperforms state-of-the-art baselines by 3.73% and 4.11% while utilizing 2x and 600x fewer samples, respectively. The implementation of AdaPlanner is available at `https://github.com/haotiansun14/AdaPlanner`.

## 1  Introduction

Large language models (LLMs) have recently emerged as versatile autonomous agents for sequential decision-making in grounded environments. Traditional decision-making methodologies like Reinforcement Learning (RL) require extensive task-specific training data and often lack the ability to generalize across tasks and environments. In contrast, LLMs are pre-trained on massive and diverse textual data, which gives them extensive world knowledge and the ability to reason over the knowledge. This makes them highly versatile and able to handle complex, real-world scenarios that may involve multiple steps of planning and decision-making.

Existing methods that leverage LLMs as autonomous agents for decision-making can be briefly categorized into two groups (Table 1): open-loop systems and closed-loop systems. Open-loop methods [24, 27, 5, 18, 12, 17, 16] rely on pre-determined plans to accomplish the desired task without any feedback adaptation mechanism. On the other hand, closed-loop systems [25, 6, 8, 22, 9, 19, 23] incorporate environment feedback to continuously monitor system behaviors and make refinements and adjustments of the plans accordingly, which therefore is more flexible.

However, both existing open-loop and closed-loop LLM agents have inherent drawbacks. Open-loop systems are computationally cheap and simple; however, they do not consider feedback from the environment and stick to the initial plan, which lack of adaptability, and, thus, can easily generate suboptimal plans. On the other hand, most existing closed-loop methods generate a fixed plan and only update their executing actions upon environment feedback. This causes them to make

---

[*]These authors contributed equally to this work.

37th Conference on Neural Information Processing Systems (NeurIPS 2023).

| Methods | Feedback Utilization | Instruction Type | Prompting | Decomposition | Experience Refinement |
|---|---|---|---|---|---|
| *Open-Loop Methods* | | | | | |
| CoT [24] | - | Prompting | Language | - | - |
| Least-To-Most [27] | - | Prompting | Language | Sub-Goals | - |
| Zero-Shot Planner [5] | - | Prompting | Language | - | - |
| HuggingGPT [18] | - | Prompting | Language | Sub-Goals | - |
| Chameleon [12] | - | Prompting | Language | Sub-Goals | - |
| *Implicit Closed-Loop Methods with Fixed Plan* | | | | | |
| ReAct [25] | Taking Action | Prompting | Language | - | - |
| Inner Monologue [6] | Taking Action | Prompting | Language | - | - |
| RCI [8] | Taking Action | Prompting | Language | - | - |
| ProgPrompt [22] | Taking Action | Prompting | Code | - | - |
| Code as Policies [9] | Taking Action | Prompting | Code | - | - |
| Reflexion [19] | Taking Action | Prompting | Language | - | Past Failure |
| *Explicit Closed-Loop Methods with Plan Refinement* | | | | | |
| DEPS [23] | Modifying Plan | Prompting & Training | Language | Sub-Goals | Past Failure |
| AdaPlanner | Action & Plan | Prompting | Code | Sub-Goals | Past Failure & Success |

Table 1: A comparison of methods that leverage LLMs for decision making. Each method's features are reported across five categories: 1) Environment Feedback Utilization: The method can use feedback to decide the next action (Taking Action), revise the entire plan (Modifying Plan), or do both (Action & Plan). 2) Instruction Type: The method may require prompting, training, or both. 3) Prompting Style: The method can employ either natural language or code for its planning backend. 4) Task Decomposition: The method might decompose the task into sub-goals or not. 5) Experience Refinement: The method can learn from past failure, past success, or both. The AdaPlanner proposed in this paper is highlighted in gray.

sub-optimal decisions that adapt to the environment in the short term but could have detrimental implications for future steps. DEPS [23] is the only exception, a method that modifies its entire plan based on feedback from the environment. However, it requires training a plan selector to choose the most successful plan, which requires a significant amount of task-specific data. As a result, applying this method to different tasks can be challenging.

To address the limitations of existing LLM agents, we propose AdaPlanner, a closed-loop planning method with LLM playing two roles – planner and refiner. The planner decomposes the task into manageable sub-goals and predicts environmental feedback for each. During execution, the refiner distinguishes and responds to two types of environment feedback – *in-plan feedback* is the environmental observation that aligns with the prediction, and *out-of-plan feedback* is one that deviates from the prediction. For in-plan feedback, the refiner can dynamically query the LLM to perform reasoning and extract key information from in-plan feedback expressed in natural language. This is achieved through a specific action called `ask_LLM()`, in which the LLM separately parses the observation and obtains information pertinent to subsequent actions. For out-of-plan feedback, the refiner proactively revises the entire plan and resumes to solve the current task from an intermediate point. AdaPlanner's adaptive closed-loop framework alleviates the need for prior knowledge about the feedback structure and permits the agent to instantly adopt a refined plan rather than restarting from scratch in a reset episode. This leads to a more efficient and adaptive decision-making process.

AdaPlanner operates solely via prompting, eliminating the need for a dedicated training phase and reducing its computational cost. Furthermore, AdaPlanner leverages a code-based prompting for precise planning and refinement. The use of code prompts facilitates task decomposition into sub-goals and mitigates LLM hallucination during the decision-making process. AdaPlanner also features a skill discovery process, which accumulates successful experiences to guide future planning. This feature further enhances its long-term planning ability and sample efficiency.

We formally define the planning problem with LLM, and introduce open-loop vs. closed-loop control system, which will motivate our method, in Section 2. Each component of the proposed AdaPlanner is specified in Section 3, including code-based prompting in Section 3.1, closed-loop adaptation in Section 3.2, and skill discovery in Section 3.3, and empirically justified in Section 4. The superior performance of AdaPlanner on both ALFWorld and MiniWoB++ demonstrates our proposed adaptive closed-loop framework can effectively enhance planning performance, even when faced with a limited number of samples.

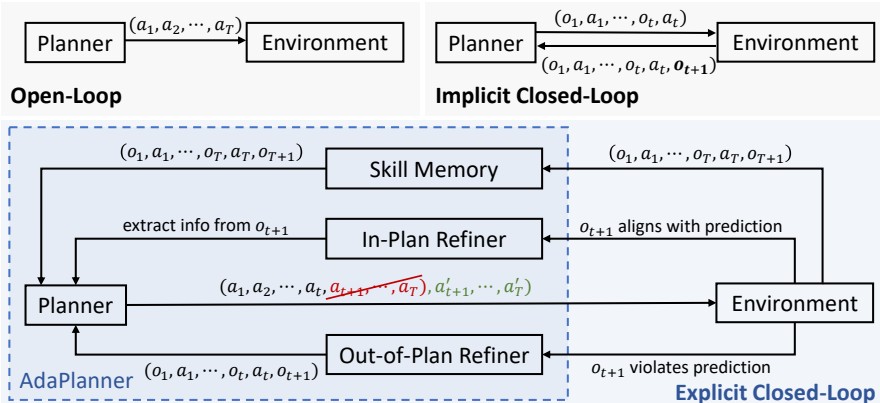

Figure 1: A comparison between open-loop, implicit closed-loop, and explicit closed-loop systems.

## 2 Preliminaries

**Problem Formulation.** We consider adopting an LLM as an autonomous agent to solve different tasks in text-based environments. For initialization, the agent is provided with allowed actions $\mathcal{A}$ in the environment, as well as a text-grounded task definition $g \in \mathcal{G}$ from the task space $\mathcal{G}$. Besides, the initial state of the environment is also observed as $o_1 \in \mathcal{O}$ from the observation space $\mathcal{O}$. With such inputs, the LLM agent needs first to generate an initial planning policy for solving the task $\rho_0(P_0|g, o_1) : \mathcal{G} \times \mathcal{O} \rightarrow \Delta(\mathcal{A}^T)$, where $T$ is the total length of steps in the generated plan and $\Delta(\cdot)$ is probability simplex function. Also, the agent can interact with the environment for feedback: When the agent interacts with the environment at the $t$-th step, the agent receives an observation $o_t \in \mathcal{O}$ from the environment and generates a trajectory-like context $c_t = (o_1, a'_1, o_2, a'_2, \cdots, a'_{t-1}, o_t)$, where $a'_1, a'_2, \cdots, a'_{t-1}$ are the executed actions within the environment. As the agent may modify the actions according to the feedback, the executed actions $a'_1, a'_2, \cdots, a'_{t-1}$ can be different from the actions $a_1, a_2, \cdots, a_{t-1}$ in the initial plan. We denote $\rho(\cdot|g, c_t, P_t)$ as the high-level planning policy that generates an entire plan and $\pi(\cdot|g, c_t, P_t)$ as the action-generation policy conditioned on a given plan $P_t$. Given the context $c_t$ and the entire plan at the last step $P_{t-1}$, the agent refines future decisions. In the end, the LLM agent should model both the initial planning policy and the environment feedback-conditioned policy to complete the given task successfully.

**Open-Loop System.** An open-loop system is a non-feedback system (Figure 1), where the output is solely dependent on the input, without any consideration of the environmental feedback. Thus, in an open-loop system, the entire initial plan over the time horizon $T$ is predetermined and static by the initial planning policy $\rho_0(\cdot|g, o_1)$, without any feedback-based refinement. Despite their simplicity, open-loop systems are notably vulnerable to environmental changes, as they lack the capacity to adapt or adjust their plans based on environmental feedback.

**Closed-Loop Systems.** On the contrary, a closed-loop system (Figure 1) refers to a planning process that incorporates environment feedback to adjust and refine future decisions, involving both initial planning $\rho_0(\cdot|g, o_1)$ and two levels of feedback-based refinements, $\rho(\cdot|g, c_t, P_{t-1})$ and $\pi(\cdot|g, c_t, P_{t-1})$, in the system.

*Implicit Closed-Loop Systems.* After each step of interaction with the environment, implicit closed-loop systems will maintain the initial plan (i.e., $P_t = P_0$) and only modify a single action based on the feedback. Therefore, the feedback-based refinement is defined as $\pi(a'_t|g, c_t, P_0)$, where $a'_t \in \mathcal{A}$ is the modified action from action space, while the remaining actions $a_{>t}$ for future steps remain the same as the initial plan. Although locally-optimal actions are adopted at each step, inaccuracies in initial planning can result in task failure or non-completion.

*Explicit Closed-Loop Systems.* Explicit closed-loop systems refine the entire plan based on environment feedback following the policy $\rho(P_t|g, c_t, P_{t-1})$, where $P_t \in \Delta(\mathcal{A}^{T-t})$ is the refined plan at time step $t$ containing the modified future actions $a'_{\geq t}$ to execute and $P_{t-1}$ is the old plan modified in the previous time step. Allowing for constant refinement and improvement of the plan, explicit closed-loop systems can help prevent costly mistakes or missed opportunities that might arise from adhering to outdated plans. Our proposed AdaPlanner is an explicit closed-loop system.

# 3 AdaPlanner

**Model Architecture.** Our AdaPlanner model, shown in Figure 1, consists of two main components:

- an LLM-based agent that functions dually as a planner and a plan refiner, and
- a skill memory module designed to enhance sample efficiency through skill discovery.

The LLM-based agent, in its planner role, generates a comprehensive plan and performs preliminary assessments to determine its feasibility. The initial planning is modeled as $\rho_0(P_0|g, o_1)$. As the plan unfolds, the agent also operates as a refiner, conducting feedback-based refinement in both in-plan and out-of-plan manners. In-plan and out-of-plan refinement processes primarily differ in how they impact future actions. In-plan refinement is a one-step action that integrates useful information into the existing plan for better action grounding. After this in-plan phase, future actions will be generated using the updated context $\pi(a'_{>t}|g, c_{>t} \cup \{h_t\}, P_0)$, where $h_t$ represents the new information obtained from $c_t$ via in-plan refinement at timestep $t$. Out-of-plan refinement, on the other hand, leverages environmental feedback to directly revise the entire plan, denoted as $\rho(P_t|g, c_t, P_{t-1})$. This mechanism allows for comprehensive adjustments to be made to the plan in response to unexpected environmental feedback. Skill memory serves as a repository, archiving past successful plans and their respective interactions with the environment. If the agent encounters a task resembling the skills stored in memory, these skills can serve as few-shot exemplars in the LLM agent's prompt. This feature improves not only sample efficiency but also reliability for future planning.

**Environment Interaction.** AdaPlanner employs adaptive closed-loop planning and active environment interaction for task solving. It can anticipate environmental observations and proactively refine the plan only when there is a discrepancy between expected and actual outcomes. This is achieved by decomposing the planning process into $N$ manageable sub-goals. During the planning and action-taking process, the agent selects from a set of timestamps, $\{t_1, \ldots, t_N\}$, to evaluate the success of each sub-goal. If the sub-goal does not align with the planned prediction at timestep $t \in \{t_1, \ldots, t_N\}$, the environment actively sends the previous sub-trajectories $(o_1, a'_1, \cdots, o_t, a'_t, o_{t+1})$ back to the refiner for plan revision. This process allows the agent to check the success status only at $N$ crucial points, thereby reducing computational costs (number of API calls) and enhancing efficiency.

## 3.1 Plan Generation via Code-Based LLM Prompting

AdaPlanner plans and refines by using Pythonic code prompts for LLMs. Consistent with previous observations [3, 2], we have found that using code prompts instead of natural language prompts for LLMs reduces ambiguity and misinterpretation, which significantly reduces LLM hallucination during plan generation and refinement. We design code prompts during different stages of decision-making, including adaptive planning, feedback generation, and in-episode refinement. We provide a detailed description of the prompts used at each stage in Appendix D.

To generate an initial plan for solving a given task, we input a task description, the permissible actions in the environment, and, when available, sample demonstrations of task resolution into LLM. These pieces of information are all formatted into Pythonic code format for LLM prompting. Figure 2 (a) shows an example programming-based plan generated by AdaPlanner for solving a `put` task in the ALFWorld environment. The generated solution function is provided with two input arguments: the first is the `agent` object, which encapsulates environmental information to be used by the agent. The second is the variable `start_from`, which is a parameter indicating the subgoal from which the agent will later resume its execution with a refined plan. By default, the `start_from` is initialized as 1. The value of this variable can be further reassigned during the refinement.

When prompting LLM to generate the code-based plan, we design the prompt to teach LLM to decompose a complex task into sub-goals. As shown in Figure 2(a), the generated code plan `solution(agent, start_from=1)` consists of: 1) a general plan at the outset that decomposes the task into subgoals in the form of comments; and 2) a sequence of sub-plans, each consisting of admissible actions corresponding to a specific subgoal. Such a mechanism allows our method to handle complex, long-horizon tasks by hierarchically decomposing them into sequences of subgoals. Furthermore, each subgoal ends with an assertion statement to test its fulfillment, which allows our method to interact actively with the environment and later resume its execution with a refined plan.

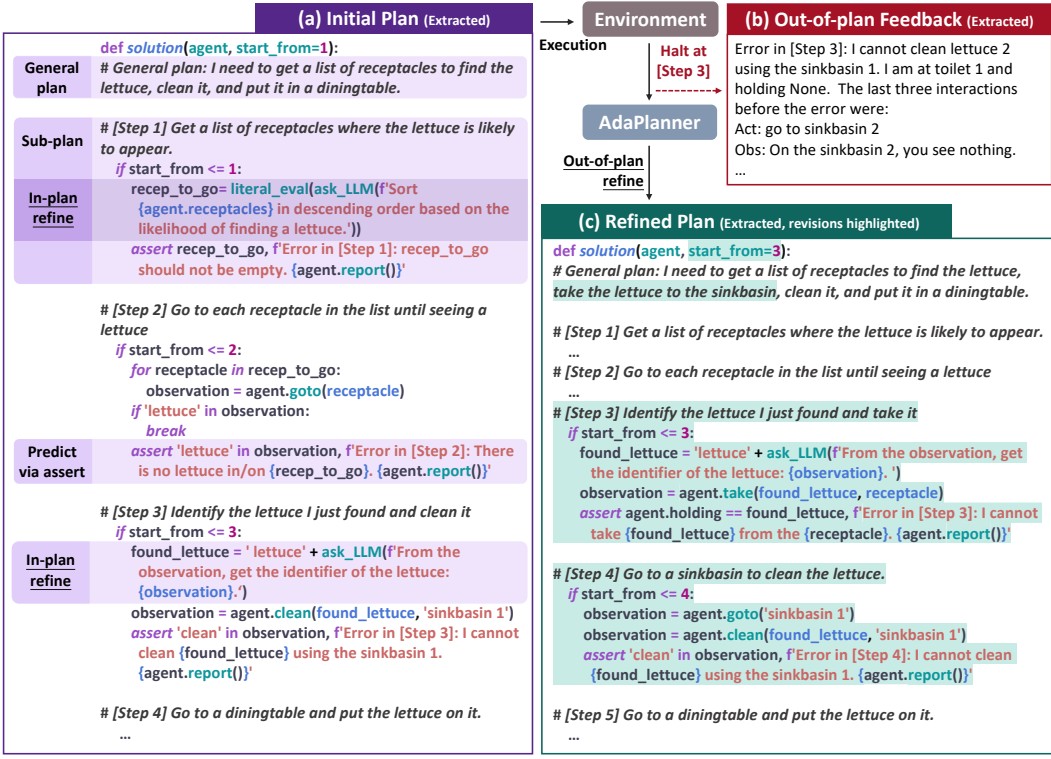

Figure 2: An illustrative example from ALFWorld to show the proposed adaptive closed-loop planning through code. The task is to `put some clean lettuce on the diningtable`. The *in-plan feedback* in (a) is a sentence like `On the countertop 2, you see a knife 1, a lettuce 1, a saltshaker 2, and a soapbottle 1`. This feedback is managed by the `ask_LLM()` action. The execution of the initial plan might yield misaligned observations, triggering an out-of-plan feedback and refinement process. For instance, the agent cannot clean the `lettuce` if it is not currently located at a `sinkbasin`. The *out-of-plan feedback* in (b) assists AdaPlanner in generating a revised plan (c) so that the agent will move to a `sinkbasin` before cleaning the `lettuce`. AdaPlanner then determines to resume from step 3 within the same episode. The task can be successfully completed using the refined plan.

## 3.2 Adaptive Closed-Loop Plan Refinement

Once an initial plan is generated, AdaPlanner then prompts the LLM to correct any syntax errors. After this, the code undergoes execution through the environment interface. The interface is responsible for grounding the actions in the environment, and also for routing environmental observations back to the code as a return value. This bi-directional flow allows AdaPlanner to adapt and refine its plan in response to environmental observations in a closed-loop manner.

**In-Plan Feedback and Refinement via `ask_LLM()` Action.** When AdaPlanner observes that the environment is aligned with the anticipated plan, it performs in-plan refinement. This allows it to extract useful information from the observation that can be used for upcoming actions. To achieve this, we provide the agent with an additional action called `ask_LLM()`, which is used to formulate a plan alongside task-specific actions. The `ask_LLM()` function enables AdaPlanner to self-query and perform reasoning based on specific information parsed from environmental observations. For instance, in [`Step 3`] in Figure 2 (a), the `ask_LLM()` action extracts the identifier of the found object `lettuce` from the natural-language observation. This information can then be fed into later actions. As an additional atomic action, this in-plan refinement is integrated into the plan at any point where the planner deems a reasoning process is necessary. Existing code-generation-based methods [22, 9, 3] face a challenge in this task, especially when there is no prior knowledge of the structure and organization of these feedback sentences. In contrast, our AdaPlanner method leverages LLM to parse critical information from diverse feedback presented in natural-language sentences to streamline plan execution.

**Out-of-Plan Refinement with the Refine-Then-Resume Mechanism.** After each sub-plan execution, AdaPlanner actively checks an assertion condition to ensure that the current plan is proceeding as expected. If the assertion fails, AdaPlanner performs out-of-plan refinement. For example, in Figure 2 (a), after `[Step 3]`, the agent is expected to hold `lettuce`. If this condition is not met, AdaPlanner generates an error message that details the current progress of execution gathered by the `report()` function. In ALFWorld tasks, this function provides a report of the agent's location, the object it is holding, and the last three interactions with the environment, as shown in Figure 2 (b). AdaPlanner then utilizes this information to perform out-of-plan refinement.

During the out-of-plan refinement as in Figure 2 (c), AdaPlanner uses a prompt similar to the one used during the initial planning stage, but with an additional feedback message that reflects the current state. Detailed prompts are provided in Appendix D. AdaPlanner then refines the plan based on the newly acquired information and also determines the value of `start_from` by comparing the plan before and after the refinement. The newly refined `solution()` is then executed from the breakpoint `start_from`. This breakpoint contains all variable states that were saved prior to refinement. Consequently, the current episode can continue from an intermediate checkpoint without restarting from scratch. We call this mechanism *refine-then-resume*. It significantly speeds up task completion and reduces the number of LLM calls required.

### 3.3 Skill Discovery

Acquiring expert demonstrations for task solving can be costly, particularly as the number of tasks increases. To address this issue, we have equipped AdaPlanner with a skill discovery feature. This is a memory scheme that discovers and archives successful trajectories, thereby improving planning performance when dealing with similar tasks. The skill discovery process consists of two stages, which can be conducted alternately over several rounds, based on the interaction costs and computation resources.

**Skill Acquisition.** In the first stage, AdaPlanner attempts to solve unseen tasks, leveraging a limited number of human demonstrations of other simpler tasks, or even no demonstrations. The model capitalizes on adaptive closed-loop planning to iteratively explore and refine solutions via a trial-and-error approach. Upon successful completion of a given task, the latest solution and the corresponding interactions are treated as candidate discovered skills.

**Skill Filtering.** In the second stage, we compare the planning performance with and without the integration of the discovered solution into the prompt. If the inclusion of this solution boosts the success rate, it is archived as a discovered skill. Conversely, if it does not improve performance, it is discarded. This filtering stage is crucial because the iterative closed-loop refinement may integrate episode-specific information into the revised solution, potentially compromising its generalizability.

## 4 Evaluation

We test AdaPlanner on two text-based decision-making environments: 1) **ALFWorld** [20] is a text-based virtual household environment encompassing six distinct task types set. We evaluate AdaPlanner on a total of 134 tasks across these six types. 2) **MiniWoB++** [11] is a simulation environment that covers a large range of computer tasks. We select 9 MiniWoB++ tasks with environmental feedback, and we also adopt and test the 53 tasks evaluated in RCI [8]. Both environments aim to solve complicated challenges with long-horizon solutions and sparse rewards. We also carefully designed ablation studies to justify the significance of each component in AdaPlanner. The Setup details and prompts for AdaPlanner are depicted in Appendix A and D. Detailed introductions to each baseline are presented in Appendix B Note that we evaluate different baselines for these two benchmarks. These methods utilize task-specific samples for prompting or training purposes, thus necessitating separate evaluations for each benchmark.

**Metrics.** Consistent with previous works [20, 25, 19, 7, 4, 11, 8], we use success rate (%) to evaluate the performance of tested methods. The success rate is defined as the number of successful episodes over the total number of episodes. Note that in ALFWorld, failure of an episode occurs when the total number of actions attains 50, with the task still unsolved. In MiniWoB++, failures can occur in two scenarios: either due to the execution of invalid actions or if the task remains unfinished following the execution of the entire plan.

**Main Results.** AdaPlanner consistently outperforms the existing baselines, achieving state-of-the-art performance, *i.e.*, an overall success rate of 91.79% in ALFWorld tasks (Table 2) and 91.11% in

| Method | Pick | Clean | Heat | Cool | Examine | Pick two | All (134 tasks) |
|---|---|---|---|---|---|---|---|
| *Training-Based Methods* | | | | | | | |
| BUTLER [20] | 46.00 | 39.00 | 74.00 | **100.00** | 22.00 | 24.00 | 37.00 |
| *Implicit Closed-Loop Methods with Fixed Plan* | | | | | | | |
| ReAct [25] (GPT-3) | 66.67 | 41.94 | 91.03 | 80.95 | 55.56 | 35.29 | 61.94 |
| ReAct [25] (GPT-3.5) | 37.50 | 64.52 | 69.57 | 42.86 | 38.89 | 17.65 | 47.76 |
| Reflexion [19] (GPT-3 + 3.5) | 75.00 | 90.32 | 91.30 | 90.48 | 88.89 | **94.12** | 88.06 |
| Reflexion [19] (GPT-3.5) | 50.00 | 41.94 | 65.22 | 52.38 | 66.67 | 47.06 | 52.99 |
| *Explicit Closed-Loop Methods with Plan Refinement* | | | | | | | |
| AdaPlanner (GPT-3) | **100.00** | **96.77** | **95.65** | **100.00** | **100.00** | 47.06 | **91.79** |
| AdaPlanner (GPT-3.5) | 77.78 | 93.55 | 69.57 | 93.65 | 62.96 | 78.43 | 80.60 |

Table 2: Success rate (%) of tested methods on six ALFWorld tasks. For ReAct and AdaPlanner, GPT-3.5 refers to `gpt-3.5-turbo`, while GPT-3 represents `text-davinci-002`. For Reflexion, GPT-3.5 indicates `gpt-3.5-turbo`. GPT-3+3.5 is used in the original Reflexion implementation, which utilizes both GPT-3 (`text-davinci-002`) and GPT-3.5 (`text-davinci-003`) for action generation and failure reflection, respectively. AdaPlanner is prompted with one specific example per task, making up six demonstrations in total. This is *half the number of samples* used in React and Reflection. The best-performing results are marked in bold. The results of our method are colored in gray.

MiniWoB++ tasks with feedback (Table 3). Specifically, in ALFWorld, AdaPlanner equipped with GPT-3 achieves a remarkable success rate exceeding 95% in the majority of individual tasks. It also surpasses all other baselines in the `Pick`, `Clean`, and `Examine` tasks. Notably, even in the task with the lowest performance (`Pick two`), AdaPlanner still outperforms BUTLER and ReAct. In the MiniWoB++ environment, AdaPlanner demonstrates superiority over all other methods on tasks that provide feedback. This superior performance suggests that AdaPlanner effectively leverages feedback to refine its plans and enhance its performance. Furthermore, AdaPlanner maintains competitive performance on tasks without feedback, achieving a success rate of 93.22%. Note that AdaPlanner's success rates of tasks without feedback are still comparable to CC-Net, the state-of-the-art model requiring over 23,000 samples per task. This result highlights the efficacy of the programming-based planning strategy employed by AdaPlanner. In both environments, AdaPlanner consistently delivers superior or competitive performance when compared to not only training-based methods but also implicit closed-loop methods under the same LLM models. These results affirm the effectiveness of the proposed explicit closed-loop plan refinement in AdaPlanner.

Furthermore, we summarize the relationship between success rate (%) and the number of samples in Figure 3. In ALFWorld, AdaPlanner yields the highest performance with the fewest number of samples. In MiniWoB++, our method outperforms most baselines. Notably, our method achieves performance comparable to CC-Net but requires 600 times fewer samples. This study highlights that AdaPlanner significantly reduces the need for extensive demonstrations or expert trajectories, thereby offering a more resource-efficient solution.

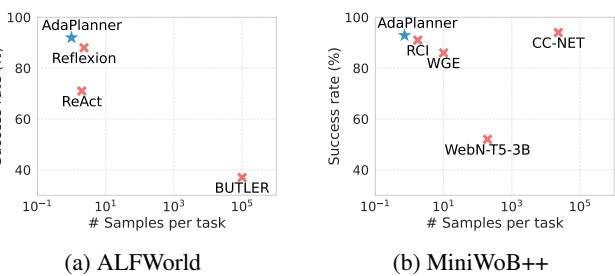

(a) ALFWorld      (b) MiniWoB++

Figure 3: Relationship between success rate (%) and the number of expert demonstrations in ALFWorld and MiniWoB++ environments. We adopt the same settings as in Table 2 (GPT-3 version) and Table 3. The top-left corner represents the pinnacle of sample efficiency.

**Adaptive Closed-Loop Architecture Enhances Planning Performance.** Figure 4a shows the performance v.s. the number of closed-loop refinements, under settings with different numbers of demo samples. The detailed example selection for this study is provided in Appendix A. We observe a significant trend of increased success rates corresponding to each subsequent closed-loop plan refinement. This indicates the AdaPlanner's ability to consistently leverage real-time feedback for performance enhancement, regardless of the number of samples used. Remarkably, AdaPlanner

| Method | With feedback (9 tasks) | No feedback (44 tasks) | All (53 tasks) |
|---|---|---|---|
| *Training-Based Methods* | | | |
| CC-Net [7] | 87.00 | **95.66** | **94.00** |
| WGE [11] | 67.60 | 87.93 | 86.00 |
| *Finetuning-Based Methods* | | | |
| WebN-T5-3B [4] | 38.50 | 54.67 | 52.00 |
| *Implicit Closed-Loop Methods with Fixed Plan* | | | |
| RCI [8] | 81.56 | 92.68 | 91.00 |
| *Explicit Closed-Loop Methods with Plan Refinement* | | | |
| AdaPlanner | **91.11** | 93.22 | 92.87 |

Table 3: Success rate (%) of tested methods on two subsets of tasks in the MiniWoB++ environment. RCI and AdaPlanner harness GPT-3.5 (`gpt-3.5-turbo` and `text-davinci-003`) as backends. Our AdaPlanner method is provided with 38 human-written demonstrations; then, it automatically discovers 21 additional examples via skill discovery, which makes up the final set of 59 examples for 53 tasks. This is *around half the number of samples* used in RCI and *over one six hundredths of the number of samples* used in CC-Net. The best-performing results are marked in bold. The results of our AdaPlanner are colored in gray. Per-task success rates are provided in Appendix F.

maintains this trend of success rate enhancement even when the total number of demonstrations across all six tasks is as low as two. In addition, as displayed in Table 4, AdaPlanner consistently outperforms both ReAct and Reflexion in producing the shortest trajectory lengths across all tasks in ALFWorld. Moreover, a comparison with Reflexion, depicted in Figure 4b, shows AdaPlanner's consistently superior performance across all iterations of closed-loop corrections. These observations highlight AdaPlanner's sample efficiency and its potential for real-world applications where the number of available demonstrations is limited.

| Method | Pick | Clean | Heat | Cool | Examine | Pick two | All (134 tasks) |
|---|---|---|---|---|---|---|---|
| ReAct[25] | 19.55 | 25.79 | 19.7 | 27.86 | 29.72 | 36.29 | 25.81 |
| Reflexion[19] | 18.77 | 25.25 | 19.13 | 25.76 | 27.11 | 35.76 | 18.90 |
| AdaPlanner | **10.79** | **13.45** | **17.61** | **13.33** | **21.00** | **20.71** | **15.60** |

Table 4: Average trajectory length (# steps) per task for ReAct, Reflexion, and AdaPlanner. We adopt the same settings as in Table 2 (GPT-3 version). Following the setting in [25, 19], each episode will be terminated after reaching 50 steps.

**Code Interface Mitigates Hallucination.** The latest `gpt-3.5-turbo` is reported to be the most capable GPT-3.5 model while reducing the cost by a tenth compared to other prevailing GPT-3 [1] and 3.5 models [14] (*e.g.*, `text-davinci-002` and `text-davinci-003`.) However, our findings from Table 2 indicate that `gpt-3.5-turbo` underperforms in decision-making tasks relative to its predecessors, i.e., `text-davinci-002`, in all LLM-agents. Upon examination of trajectories from both models, we observed a noticeable hallucination with GPT-3.5 (`gpt-3.5-turbo`), as shown in Appendix E. We hypothesize that `gpt-3.5-turbo` might be a smaller-scale model that is more prone to hallucination. We also found a similar hypothesis drawn based on various experiments in [26]. Furthermore, `gpt-3.5-turbo` is primarily optimized for human conversation tasks, which could potentially compromise its performance on tasks such as code generation and reasoning[13]. Despite this, AdaPlanner demonstrates a remarkable level of resilience against hallucination even with `gpt-3.5-turbo` (Table 2), while ReAct and Reflexion are more sensitive to the hallucination issue. AdaPlanner's resilience against hallucination can be attributed to its use of code prompts, which provide a more formal and constrained generation space for LLM. For comparison, we implement an ablation version of AdaPlanner without the code interface by translating solution examples directly into plans and actions using natural language. Without the code interface, AdaPlanner's performance substantially drops in both ALFWorld and MiniWoB++ environments (Figure 4c), from 81% to 46% and from 93% to 66%, respectively. This significant performance drop underscores the essential role of the code interface in AdaPlanner.

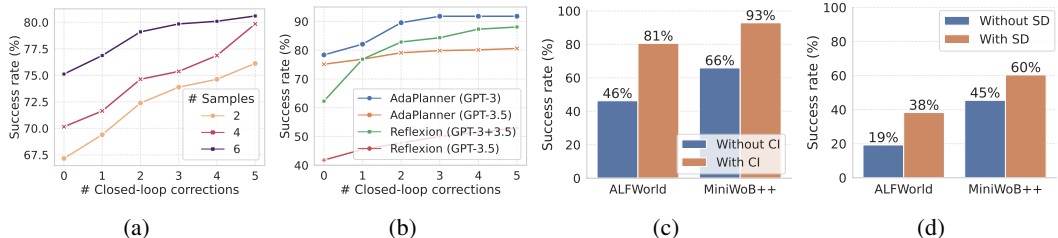

Figure 4: Performance comparison on 134 ALFWorld tasks in different cases. We adopt the same settings as in Table 2. (a) and (b) presents the success rate (%) with different numbers of closed-loop corrections: (a) compares AdaPlanner with different numbers of samples; (b) compares AdaPlanner and Reflexion with two LLMs. (c) shows the success rate (%) of AdaPlanner with and without code interface (CI). (d) shows the success rate (%) of AdaPlanner with and without skill discovery (SD). Note that for (a), the number signifies the total number of samples used across all six tasks.

**Skill Discovery Improves Sample Efficiency.** The skill discovery in AdaPlanner utilizes a long-term memory mechanism that retains successful solutions, thus boosting planning performance when faced with similar tasks. An ablation study depicted in Figure 4d compares the performance of AdaPlanner with and without the implementation of skill discovery. In the skill acquisition stage, we provide a maximum of one demonstration. In ALFWorld, AdaPlanner is prompted with only one expert demonstration of the simplest task (put). We evaluate the average success rate of the method on the remaining five tasks, which are comparatively more challenging and require additional steps for completion. In MiniWoB++, we apply zero-shot prompting, omitting any examples in the skill acquisition phase. For both environments, we operate the method using GPT-3.5 in adaptive closed-loop mode, and one round of skill discovery is conducted. As Figure 4d illustrates, the inclusion of skill discovery significantly enhances performance. In the ALFWorld tasks, the success rate of AdaPlanner nearly doubles when skill discovery is employed. Similarly, in the MiniWoB++ tasks, the overall success rate increases by approximately 15% with skill discovery. Moreover, we conducted an ablation study on MiniWoB++ to evaluate the impact of sample numbers, as detailed in Table 5. AdaPlanner with skill discovery requires only 15 samples to outperform the variant without skill discovery, even though the latter used twice as many samples. It is evident that skill discovery enhances sample efficiency.

| # samples | 38 | 30 | 20 | 15 | 0 |
|---|---|---|---|---|---|
| With SD | 92.87 | 84.06 | 79.17 | 75.17 | 60.38 |
| Without SD | 82.40 | 73.58 | 68.70 | 64.70 | 45.47 |

Table 5: Success rate (%) of AdaPlanner on MiniWob++ with and without skill discovery (SD) across varying numbers of expert samples per task. We adopt the same settings as in Table 2 (GPT-3 version) and Table 3.

## 5 Related Work

Many works have studied how to leverage LLMs as autonomous agents to accomplish decision-making tasks within text-based environments. Earlier studies, like Chain-of-Thoughts [24] and Zero-Shot Planner [5], utilize prompts to guide LLMs in generating complete action sequences for elementary tasks. For more complex tasks, methods like HuggingGPT [18] and Chameleon [12] also generate the initial plan of using different tools and then call the corresponding APIs for execution. Meanwhile, some other works also prompt LLMs to compose plans in Planning Domain Definition Language (PDDL) [10, 15]. However, all these plans are created in an open-loop fashion without adapting to feedback from external environments.

To address the limitations of open-loop systems, recent techniques have emerged that focus on establishing closed-loop systems. These systems are capable of leveraging environmental feedback, thereby facilitating more adaptive decision-making. ReAct [25] and Inner Monologue [6] allow LLM agents to take single-step actions according to the environmental feedback. Reflexion [19], as an extension of ReAct, tries to resolve this issue by enabling the ReAct agent to revise itself from past trials and errors. Moreover, RCI [8] starts by formulating a comprehensive plan, modifying the immediate action when the agent encounters a failure at the current step. While all the aforementioned

methods can adapt their decisions based on environmental feedback, they assume the LLM-generated initial plan is correct and do not adjust it. Rather, they solely modify the immediate action being executed and are easy to fall into local sub-optimal actions without considering the long-term plans.

To further enhance the agents' both capabilities of planning and adapting to environmental feedback, strict closed-loop architectures are proposed that can recursively refine the generated plans. DEPS [23] is one of the examples that initially proposes an entire plan and then applies real-world feedback to recursively refine it during execution. However, this method requires training a selector to generate a plan that is highly probable to succeed, which makes it difficult to generalize the plans and actions to other tasks. Besides, the required data for training the plan selector are often unavailable in practice and expensive to collect. In contrast, AdaPlanner generates and refines plans via LLM prompting, making it widely applicable to various decision-making problems. Another PDDL-based planning method is proposed in [21] with the ability to leverage feedback and modify the plans. However, the plan formulated on the training set remains static and lacks task-specific refinement during the execution of the evaluation task. In contrast, AdaPlanner dynamically refines the plan and adapts to various feedback throughout the task-solving process.

## 6  Conclusion and Limitations

We proposed AdaPlanner, a closed-loop approach enabling LLM agents to adaptively refine their generated plans according to environment feedback. We defined two different refinement strategies, in-plan and out-of-plan refinement, to fully leverage environment information. Furthermore, to mitigate the LLMs' hallucination issue and make them learn from the past experience, we proposed code-style prompting and skill discovery mechanisms. Through comprehensive experiments, we demonstrated that AdaPlanner outperforms the state-of-the-art baselines significantly and has better sample efficiency. Our ablation studies also showed the effectiveness of different components in AdaPlanner. One limitation of AdaPlanner is that it still require few-shot expert demonstrations for solving complex tasks. Although AdaPlanner has already achieved better sample efficiency than existing methods, it is interesting to study how to further enhance AdaPlanner to solve complex tasks with no demonstrations in the future.

## 7  Broader Impacts

Our research approach focuses on treating LLMs as autonomous agents and improving their ability to solve complex sequential decision-making tasks. However, this research line carries inherent risks, including security threats, potential misuse, and unintended consequences such as job displacement due to automation. To mitigate these risks, it is essential for researchers and policymakers to collaborate in creating and implementing effective regulations to guide the development and deployment of these technologies towards positive outcomes. Additionally, we believe that the research community should coordinate efforts to design principles and techniques that prioritize safety and human values before LLM agents are deployed in various industries. This will help ensure that LLMs are aligned with ethical and moral standards while promoting their positive impact on society.

## Acknowledgments and Disclosure of Funding

This work was supported in part by NSF (IIS2008334, IIS-2106961, CAREER IIS-2144338), ONR (MURI N00014-17-1-2656), IDEaS Cyberinfrastructure Resources, and Microsoft Accelerate Foundation Models Research Program.

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

## A    Experimental Setup

**ALFWorld** [20] is a comprehensive suite of synthetic, text-based environments, encompassing six distinct task types set, `Pick`, `Clean`, `Heat`, `Cool`, `Examine`, and `Pick two`, within a virtual household. Each task possesses a unique high-level objective (e.g., `put some vase in safe`, *etc.*) that necessitates agent navigation and interaction with various objects or receptacles (e.g., `go to shelf 6`, `clean apple`, *etc.*). To fulfill the stipulated task, the agent is required to implement a sequence of actions aimed at accomplishing the predetermined goal. However, given that an object may potentially reside in any one of over 50 possible locations in a task instance, the agent must sequentially explore each of these. Consequently, the entire action trajectory could involve more than 50 individual actions, presenting a significant challenge to the agent.

**MiniWoB++** [11] is a task suite of simulation environments that covers a large range of computer tasks for net agents. The computer tasks start from simple button-clicking to more challenging ones with longer time horizons (*e.g.*, `click-checkboxes-large`), reasoning (*e.g.*, `click-checkboxes-soft`), unexpected pop-ups (*e.g.*, `login-user-popup` ), and stochastically varying layouts (*e.g.*, `multi-orderings`, `multi-layouts`). These challenges are suitable for evaluating our proposed closed-loop framework. Each task interacts with a 160px × 210px web environment, where the state space is purely the HTML code of the web. Following RCI [8], we define the actions space as two sets of operations, i.e., clicking and typing actions. The clicks allow the agent to interact with clickable HTML elements (*e.g.*, webpage buttons). The typings are conducted with keyboard-based actions, such as inputting characters into the input box and stroking functional keys (*e.g.*, ENTER, BACKSPACE). We select nine MiniWoB++ tasks where environment observations (i.e., the HTML code) change after certain actions: `search-engine`, `tic-tac-toe`, `terminal`, `login-user-popup`, `guess-number`, `email-inbox`, `email-inbox-nl-turk`, `email-inbox-forward-nl`, and `email-inbox-forward-nl-turk`. Take the task `terminal` as an example. Search results appear on the webpage after inputting the keyword and pressing the search button. Therefore, environment feedback can be interpreted from the change in HTML code and further leveraged by closed-loop planning. We also adopt and test the 53 tasks evaluated in RCI [8].

## B    Baseline Details

**ALFWorld.** Following a set of previous works [20, 25, 19], we evaluate AdaPlanner on 134 different environments. By default, we include one sample as an exemplar per task to prompt AdaPlanner. For the study presented in Figure 4a, we adopt the setting of prompted samples as in Table 6. For the study in Figure 4d, we use one sample of the simplest task `put` to prompt the rest of the five tasks, which are more complex and require more steps to solve. For baselines, we compare AdaPlanner with BUTLER [20], ReAct [25], and Reflexion [19]. BUTLER [20] is an imitation learning method trained with 100k samples per task. ReAct and Reflexion, as outlined in Table 1, are prompting-based methodologies utilizing an implicit closed-loop framework. They employ a total of 6 and 8 samples, respectively, across all six tasks. BUTLER results are sourced from [20]. We evaluate ReAct, Reflexion, and AdaPlanner empowered by both GPT-3 (`text-davinci-002`) and GPT-3.5 (`gpt-3.5-turbo` and `text-davinci-003`) models.

| # samples | Pick | Clean | Heat | Cool | Examine | Pick two |
|---|---|---|---|---|---|---|
| 2 | Clean | Clean | Clean | Clean | Examine | Examine |
| 4 | Pick | Clean | Clean | Clean | Examine | Pick two |
| 6 | Pick | Clean | Heat | Cool | Examine | Pick two |

Table 6: The specific allocation of samples for prompting each task is divided into three cases based on the total number of samples (2, 4, and 6) used across the six types of tasks. For instance, when a total of 2 samples are used for all tasks, a single expert trajectory sample for the `Clean` task is utilized to prompt four tasks (`Pick`, `Clean`, `Heat`, and `Cool`). Similarly, a sample from the `Examine` task is used to prompt two tasks (`Examine` and `Pick two`).

**MiniWoB++.** Overall, we report the evaluation results of RCI [8] and the proposed AdaPlanner in GPT-3.5 (`text-davinci-003`), along with three training or finetuning-based baselines: Computer

Control Agent Architecture(CC-Net) [7], Workflow-Guided Exploration(WGE) [11], and WebN-T5-3B [4]. CC-Net and WGE employ supervised learning and reinforcement learning with over 23K and 10 demonstrations per task, respectively. WebN-T5-3B uses 12K demonstrations to finetune a pre-trained language model. RCI is a prompting-based approach that is categorized as the implicit closed-loop method in Table 1, which utilizes 93 samples across the 53 tasks. For these 53 tasks, we first provide AdaPlanner with 38 human-written demonstrations and perform skill discovery to obtain another 21 additional examples, i.e., 59 examples are used for 53 tasks. Evaluation results of RCI, CC-Net, WGE, and WebN-T5-3B are sourced from the works of [8, 7, 11, 4], respectively.

## C  Proportion of two refinements

| Environment | # In-plan refinement | # Out-of-plan | Proportion (in/out) |
|---|---|---|---|
| ALFWorld(134 tasks) | 2.83 | 6.40 | 0.44 |
| MiniWoB++(9 tasks with feedback) | 0.33 | 1.78 | 0.19 |

Table 7: Average number and ratio of in-plan and out-of-plan refinement invoked per task.

Table 7 illustrates that out-of-plan refinement is invoked more frequently in both environments, emphasizing its important role in AdaPlanner's planning process. The average occurrence for both types of refinements varies between ALFWorld and MiniWoB++. This can be attributed to the differences in the average lengths of the trajectories in ALFWorld and MiniWoB++ (15.60 and 5.40 steps, respectively).

## D  Prompts

### D.1  ALFWorld

**Basic Information.** The `<basic_info>` defines the agent and admissible actions for AdaPlanner. Note that the actual definitions of action functions are not specified in the prompt. Instead, only a formalized definition with several examples is provided, such that the planner can acquire how to compose a plan based on these actions. As can be seen in the following part, this `<basic_info>` prompt is used in both `<initial_planning>` and `<refinement>` prompts.

```
<basic_info> Prompt

# You are a household agent. Here is some Python code defining a household
environment:

# Use literal_eval to convert the answer from ask() to a list.
from ast import literal_eval

# In the environment, you can ask questions to an assistant by ask():
from large_language_model import ask_llm as ask
# for example: You have a list of receptacles, and you want to sort them by the
likelihood of a soapbar appearing in them. You can do this by asking the
assistant:
receptacles = ['countertop 1', 'garbagecan 1', 'sinkbasin 2', 'sinkbasin 1',
'toilet 1', 'toiletpaperhanger 1', 'towelholder 1']
answer = ask(f'Sort the list of receptacles, starting from the one a soapbar is
most likely to appear: {receptacles}. You should return a Python list.')
# answer = ['sinkbasin 1', 'sinkbasin 2', 'countertop 1', 'towelholder 1',
'toiletpaperhanger 1', 'garbagecan 1', 'toilet 1']

# Agent class represents the state of the agent, including its location,
# what it's holding as well as the actions it can take.
class Agent:
    def __init__(self, receptacles):
        self.location = None
        self.holding = None
        self.receptacles = receptacles
```

```
# Here are the admissible actions the agent can take:

# Go to a receptacle and update the agent's location.
# For example, 'On the countertop 1, you see a candle 1, a cloth 2, and a
soapbar 1.' = goto('countertop 1')
# For example, 'On the sidetable 2, you see nothing.' = goto('sidetable 2')
def goto(self, receptacle):
    ...

# Take an object from a receptacle if the agent is not holding anything.
# For example, 'You pick up the soapbar 1 from the towelholder 1.' =
take('soapbar 1', 'towelholder 1')
def take(self, object, receptacle):
    ...

# Put an object in or on a receptacle if the agent is holding it.
# For example, 'You put the soapbar 1 in/on the cabinet 1.' = put('soapbar
1', 'cabinet 1')
def put(self, object, receptacle):
    ...

# Open a receptacle and observe its contents.
# For example, 'You open the cabinet 1. The cabinet 1 is open. In it, you
see a cloth 1.' = open_receptacle('cabinet 1')
def open_receptacle(self, receptacle):
    ...

# Clean an object with a receptacle.
# For example, 'You clean the soapbar 1 using the sinkbasin 1.' =
clean('soapbar 1', 'sinkbasin 1')
def clean(self, object, receptacle):
    ...

# Heat an object with a receptacle.
# For example, 'You heat the tomato 1 using the microwave 1.' = heat('tomato
1', 'microwave 1')
def heat(self, object, receptacle):
    ...

# Cool an object with a receptacle.
# For example, 'You cool the pan 2 using the fridge 1.' = cool('pan 2',
'fridge 1')
def cool(self, object, receptacle):
    ...

# Turn on an object.
# For example, 'You turn on the desklamp 1.' = turn_on('desklamp 1')
def turn_on(self, object):
    ...

# Report agent's current state, including its location, what it's holding,
and last action and observation.
# This function should only be used in assertion.
def report(self):
    ...
```

**Initial Planning.** The <initial_planning> prompt is employed to generate the preliminary plan. In this context, <basic_info> is substituted by the content of the <basic_info> prompt. The <sample> is replaced with an expert trajectory, while <receptacle_list> is substituted by the list of interactive receptacles provided by the task environment. Finally, <task> is substituted by the task description, expressed in natural language.

```
┌────────── <initial_planning> Prompt ──────────┐
<basic_info>

# Now complete the function solution() below to solve the task by composing the
agent's methods to interact with the environment.
# For each step you plan to take, 1) mark with '[Step xx]', 2) give a reason why
you think it is a good step to take 3) write an assertion to check if the step
is successful.

# Here is an example of a solution to the task:

<sample>

# Here is the actual task.
# define environment and agent
receptacles = <receptacle_list>
agent = Agent(receptacles)

# <task>
# You should complete your solution function below:
def solution(agent, start_from=1):
```

**Samples.** In ALFWorld, there are six types of tasks: `Pick`, `Clean`, `Heat`, `Cool`, `Examine`, and `Pick two`. For each type, we gather one expert sample of solutions that the planner can refer to. These six expert samples are presented as follows:

The expert sample for the task `Pick`:

```
┌────────── <sample_pick> Prompt ──────────┐
# define environment and agent
receptacles = ['diningtable 1','drawer 2', 'drawer 1', 'sinkbasin 1', 'toilet
1', 'sidetable 2', 'sidetable 1', 'cabinet 1', 'countertop 1', 'microwave 1',
'fridge 1']
agent = Agent(receptacles)

# Your task is to: put soapbar on countertop.
# here is a solution:
def solution(agent, start_from=1):
    # General Plan: I need to get a list of receptacles where the soapbar is
    likely to appear, and then go to each receptacle in the list until seeing a
    soapbar. Then I can put get the identifier of the soapbar and take it.
    Finally I can go to the countertop and put the soapbar.
    if start_from <= 1:
        print("[Step 1] get a list of receptacles where the soapbar is likely to
        appear")
        # I can ask the assistant to do that.
        answer = ask(f'Given a list of receptacles, please sort them in
        descending order based on the likelihood of finding a soapbar in each of
        them. The list of receptacles is: {agent.receptacles}. You should
        directly return a Python list.')
        recep_to_check = literal_eval(answer)
        # expectation: the returned recep_to_check should not be empty.
        assert recep_to_check, f'Error in [Step 1]: recep_to_check should not be
        empty. {agent.report()}'

    if start_from <= 2:
        print("[Step 2] go to each receptacle in the list until seeing a
        soapbar")
        for receptacle in recep_to_check:
            observation = agent.goto(receptacle)
            # check if the receptacle is closed. If so, open it.
            if 'closed' in observation:
                observation = agent.open_receptacle(receptacle)
            # check if a soapbar is in/on the receptacle.
```

```
              if 'soapbar' in observation:
                  break
         # expectation: I should be able to find a receptacle where a soapbar is
         in/on it.
         assert 'soapbar' in observation, f'Error in [Step 2]: There is no
         soapbar in/on {recep_to_check}. {agent.report()}'

     if start_from <= 3:
         print("[Step 3] identify the soapbar I juts found and take it")
         # I need to get the identifier of the soapbar. I can ask the assistant
         to do that.
         answer = ask(f'From the observation, get the identifier of an object.
         For example, On the cabinet 1, you see a cloth 2, and a toiletpaper 2.
         The identifier of cloth is 2. Now, {observation} The identifier of the
         soap? Only Output a single number without any other words. ')
         found_soapbar = f'soapbar {answer}'
         observation = agent.take(found_soapbar, receptacle)
         # expectation: I should be able to take the soapbar from the receptacle.
         assert agent.holding == found_soapbar, f'Error in [Step 3]: I cannot
         take {found_soapbar} from the {receptacle}. {agent.report()}'

     if start_from <= 4:
         print("[Step 4] go to a countertop and put the soapbar on it")
         # There are multiple countertops, and I only need to go to one of them.
         observation = agent.goto('countertop 1')
         # check if the countertop is closed. If so, open it.
         if 'closed' in observation:
             observation = agent.open_receptacle('countertop 1')
         observation = agent.put(found_soapbar, 'countertop 1')
         # expectation: I should be able to put the soapbar on the countertop.
         assert f'You put the {found_soapbar} in/on the countertop 1.' in
         observation, f'Error in [Step 4]: I cannot put the {found_soapbar} on
         the countertop 1. {agent.report()}'
```

The expert sample for the task `Clean`:

```
                        <sample_clean> Prompt
# define environment and agent
receptacles = ['diningtable 1','drawer 2', 'drawer 1', 'sinkbasin 1', 'toilet
1', 'sidetable 2', 'sidetable 1', 'cabinet 1', 'countertop 1', 'microwave 1',
'fridge 1']
agent = Agent(receptacles)

# Your task is to: put a clean lettuce in diningtable / clean a lettuce and put
it in diningtable.
# here is a solution:
def solution(agent, start_from=1):
    # General plan: I need to get a list of receptacles to find the lettuce,
    take the lettuce to the sinkbasin, clean it and put it in a diningtable.
    if start_from <= 1:
        print("[Step 1] get a list of receptacles where the lettuce is likely to
        appear.")
        # I can ask the assistant to do that.
        answer = ask(f'Given a list of receptacles, please sort them in
        descending order based on the likelihood of finding a lettuce in each of
        them. The list of receptacles is: {agent.receptacles}. You should
        directly return a Python list.')
        recep_to_check = literal_eval(answer)
        # expectation: the returned recep_to_check should not be empty.
        assert recep_to_check, f'Error in [Step 1]: recep_to_check should not be
        empty. {agent.report()}'

    if start_from <= 2:
        print("[Step 2] go to each receptacle in the list until seeing a
        lettuce")
```

```
        for receptacle in recep_to_check:
            observation = agent.goto(receptacle)
            # check if the receptacle is closed. If so, open it.
            if 'closed' in observation:
                observation = agent.open_receptacle(receptacle)
            # check if a lettuce is in/on the receptacle.
            if 'lettuce' in observation:
                break
        # expectation: I should be able to find a receptacle where a lettuce is
        in/on it.
        assert 'lettuce' in observation, f'Error in [Step 2]: There is no
        lettuce in/on {recep_to_check}. {agent.report()}'

    if start_from <= 3:
        print("[Step 3] identify the lettuce I juts found and take it")
        # I need to get the identifier of the lettuce. I can ask the assistant
        to do that.
        answer = ask(f'From the observation, get the identifier of an object.
        For example, On the cabinet 1, you see a cloth 2, and a toiletpaper 2.
        The identifier of cloth is 2. Now, {observation} The identifier of the
        lettuce? Only Output a single number without any other words. ')
        found_lettuce = f'lettuce {answer}'
        observation = agent.take(found_lettuce, receptacle)
        # expectation: I should be able to take the lettuce from the receptacle.
        assert agent.holding == found_lettuce, f'Error in [Step 3]: I cannot
        take {found_lettuce} from the {receptacle}. {agent.report()}'

    if start_from <= 4:
        print("[Step 4] go to a sinkbasin to clean the lettuce. ")
        # I should go to the sinkbasin first if I want to clean the lettuce.
        observation = agent.goto('sinkbasin 1')
        # check if the sinkbasin is closed. If so, open it.
        if 'closed' in observation:
            observation = agent.open_receptacle('sinkbasin 1')
        observation = agent.clean(found_lettuce, 'sinkbasin 1')
        # expectation: I should be able to clean the lettuce.
        assert f'You clean the {found_lettuce} using the sinkbasin 1.' in
        observation, f'Error in [Step 4]: I cannot clean the {found_lettuce}
        using the sinkbasin 1. {agent.report()} I should have been at sinkbasin
        1 and holding {found_lettuce}.'

    if start_from <= 5:
        print("[Step 5] go to a diningtable and put the lettuce on it. ")
        # There are multiple diningtables, and I only need to go to one of them.
        observation = agent.goto('diningtable 1')
        # check if the diningtable is closed. If so, open it.
        if 'closed' in observation:
            observation = agent.open_receptacle('diningtable 1')
        observation = agent.put(found_lettuce, 'diningtable 1')
        # expectation: I should be able to put the lettuce on the diningtable.
        assert f'You put the {found_lettuce} in/on the diningtable 1.' in
        observation, f'Error in [Step 5]: I cannot put the {found_lettuce} on
        the diningtable 1. {agent.report()}'
```

The expert sample for the task Heat:

```
                    ┌─────────────────────────┐
──────────────────── <sample_heat> Prompt ────────────────────
                    └─────────────────────────┘
# define environment and agent
receptacles = ['diningtable 1','drawer 2', 'drawer 1', 'sinkbasin 1', 'toilet
1', 'sidetable 2', 'sidetable 1', 'cabinet 1', 'countertop 1', 'microwave 1',
'fridge 1']
agent = Agent(receptacles)

# Your task is to: put a hot lettuce in diningtable / heat some lettuce and put
it in diningtable.
```

```python
# here is a solution:
def solution(agent, start_from=1):
    # General plan: I need to get a list of receptacles to find the lettuce,
    take the lettuce to the microwave, heat it and put it in a diningtable.
    if start_from <= 1:
        print("[Step 1] get a list of receptacles where the lettuce is likely to
        appear.")
        # I can ask the assistant to do that.
        answer = ask(f'Given a list of receptacles, please sort them in
        descending order based on the likelihood of finding a lettuce in each of
        them. The list of receptacles is: {agent.receptacles}. You should
        directly return a Python list.')
        recep_to_check = literal_eval(answer)
        # expectation: the returned recep_to_check should not be empty.
        assert recep_to_check, f'Error in [Step 1]: recep_to_check should not be
        empty. {agent.report()}'

    if start_from <= 2:
        print("[Step 2] go to each receptacle in the list until seeing a
        lettuce")
        for receptacle in recep_to_check:
            observation = agent.goto(receptacle)
            # check if the receptacle is closed. If so, open it.
            if 'closed' in observation:
                observation = agent.open_receptacle(receptacle)
            # check if a lettuce is in/on the receptacle.
            if 'lettuce' in observation:
                break
        # expectation: I should be able to find a receptacle where a lettuce is
        in/on it.
        assert 'lettuce' in observation, f'Error in [Step 2]: There is no
        lettuce in/on {recep_to_check}. {agent.report()}'

    if start_from <= 3:
        print("[Step 3] identify the lettuce I juts found and take it")
        # I need to get the identifier of the lettuce. I can ask the assistant
        to do that.
        answer = ask(f'From the observation, get the identifier of an object.
        For example, On the cabinet 1, you see a cloth 2, and a toiletpaper 2.
        The identifier of cloth is 2. Now, {observation} The identifier of the
        lettuce? Only Output a single number without any other words. ')
        found_lettuce = f'lettuce {answer}'
        observation = agent.take(found_lettuce, receptacle)
        # expectation: I should be able to take the lettuce from the receptacle.
        assert agent.holding == found_lettuce, f'Error in [Step 3]: I cannot
        take {found_lettuce} from the {receptacle}. {agent.report()}'

    if start_from <= 4:
        print("[Step 4] go to a microwave to heat the lettuce")
        # I should go to a microwave to heat the lettuce.
        observation = agent.goto('microwave 1')
        # check if the microwave is closed. If so, open it.
        if 'closed' in observation:
            observation = agent.open_receptacle('microwave 1')
        observation = agent.heat(found_lettuce, 'microwave 1')
        # expectation: I should be able to heat the lettuce.
        assert f'You heat the {found_lettuce} using the microwave 1.' in
        observation, f'Error in [Step 4]: I cannot heat the {found_lettuce}
        using the microwave 1. {agent.report()} I should have been at microwave
        1 and holding {found_lettuce}. '

    if start_from <= 5:
        print("[Step 5] go to a diningtable and put the lettuce on it")
        # There are multiple diningtables, and I only need to go to one of them.
        observation = agent.goto('diningtable 1')
```

```
        # check if the diningtable is closed. If so, open it.
        if 'closed' in observation:
            observation = agent.open_receptacle('diningtable 1')
        observation = agent.put(found_lettuce, 'diningtable 1')
        # expectation: I should be able to put the lettuce on the diningtable.
        assert f'You put the {found_lettuce} in/on the diningtable 1.' in
        observation, f'Error in [Step 5]: I cannot put the {found_lettuce} on
        the diningtable 1. {agent.report()}'
```

The expert sample for the task `Cool`:

```
# define environment and agent
receptacles = ['diningtable 1','drawer 2', 'drawer 1', 'sinkbasin 1', 'toilet
1', 'sidetable 2', 'sidetable 1', 'cabinet 1', 'countertop 1', 'microwave 1',
'fridge 1']
agent = Agent(receptacles)

# Your task is to: put a cold lettuce in diningtable / cool some lettuce and put
it in diningtable.
# here is a solution:
def solution(agent, start_from=1):
    # General plan: I need to get a list of receptacles to find the lettuce,
    take the lettuce to the fridge, cool it and put it in a diningtable.
    if start_from <= 1:
        print("[Step 1] get a list of receptacles where the lettuce is likely to
        appear.")
        # I can ask the assistant to do that.
        answer = ask(f'Given a list of receptacles, please sort them in
        descending order based on the likelihood of finding a lettuce in each of
        them. The list of receptacles is: {agent.receptacles}. You should
        directly return a Python list.')
        recep_to_check = literal_eval(answer)
        # expectation: the returned recep_to_check should not be empty.
        assert recep_to_check, f'Error in [Step 1]: recep_to_check should not be
        empty. {agent.report()}'

    if start_from <= 2:
        print("[Step 2] go to each receptacle in the list until seeing a
        lettuce")
        for receptacle in recep_to_check:
            observation = agent.goto(receptacle)
            # check if the receptacle is closed. If so, open it.
            if 'closed' in observation:
                observation = agent.open_receptacle(receptacle)
            # check if a lettuce is in/on the receptacle.
            if 'lettuce' in observation:
                break
        # expectation: I should be able to find a receptacle where a lettuce is
        in/on it.
        assert 'lettuce' in observation, f'Error in [Step 2]: There is no
        lettuce in/on {recep_to_check}. {agent.report()}'

    if start_from <= 3:
        print("[Step 3] identify the lettuce I juts found and take it")
        # I need to get the identifier of the lettuce. I can ask the assistant
        to do that.
        answer = ask(f'From the observation, get the identifier of an object.
        For example, On the cabinet 1, you see a cloth 2, and a toiletpaper 2.
        The identifier of cloth is 2. Now, {observation} The identifier of the
        lettuce? Only Output a single number without any other words. ')
        found_lettuce = f'lettuce {answer}'
        observation = agent.take(found_lettuce, receptacle)
        # expectation: I should be able to take the lettuce from the receptacle.
```

```
        assert agent.holding == found_lettuce, f'Error in [Step 3]: I cannot
        take {found_lettuce} from the {receptacle}. {agent.report()}'

    if start_from <= 4:
        print("[Step 4] go to a fridge to cool the lettuce")
        # I should go to a fridge to cool the lettuce.
        observation = agent.goto('fridge 1')
        # check if the fridge is closed. If so, open it.
        if 'closed' in observation:
            observation = agent.open_receptacle('fridge 1')
        observation = agent.cool(found_lettuce, 'fridge 1')
        # expectation: I should be able to cool the lettuce.
        assert f'You cool the {found_lettuce} using the fridge 1.' in
        observation, f'Error in [Step 4]: I cannot cool the {found_lettuce}
        using the fridge 1. {agent.report()} I should have been at fridge 1 and
        holding {found_lettuce}.'

    if start_from <= 5:
        print("[Step 5] go to a diningtable and put the lettuce on it")
        # There are multiple diningtables, and I only need to go to one of them.
        observation = agent.goto('diningtable 1')
        # check if the diningtable is closed. If so, open it.
        if 'closed' in observation:
            observation = agent.open_receptacle('diningtable 1')
        observation = agent.put(found_lettuce, 'diningtable 1')
        # expectation: I should be able to put the lettuce on the diningtable.
        assert f'You put the {found_lettuce} in/on the diningtable 1.' in
        observation, f'Error in [Step 5]: I cannot put the {found_lettuce} on
        the diningtable 1. {agent.report()}'
```

The expert sample for the task Examine:

```
                           <sample_examine> Prompt
# define environment and agent
receptacles = ['diningtable 1','drawer 2', 'drawer 1', 'sinkbasin 1', 'toilet
1', 'sidetable 2', 'sidetable 1', 'cabinet 1', 'countertop 1', 'microwave 1',
'fridge 1']
agent = Agent(receptacles)

# Your task is to: look at the bowl under the desklamp / examine the bowl with
the desklamp
# here is a solution:
def solution(agent, start_from=1):
    # General plan: I need to get a list of receptacles to find the bowl and
    take the bowl with me, then I get another list of receptacles to find the
    desklamp and turn it on.
    if start_from <= 1:
        print("[Step 1] get a list of receptacles where a bowl is likely to
        appear.")
        # I can ask the assistant to do that.
        answer = ask(f'Given a list of receptacles, please sort them in
        descending order based on the likelihood of finding a bowl in each of
        them. The list of receptacles is: {agent.receptacles}. You should
        directly return a Python list.')
        recep_to_check = literal_eval(answer)
        # expectation: the returned recep_to_check should not be empty.
        assert recep_to_check, f'Error in [Step 1]: recep_to_check should not be
        empty. {agent.report()}'

    if start_from <= 2:
        print("[Step 2] go to each receptacle in the list until seeing a pen")
        for receptacle in recep_to_check:
            observation = agent.goto(receptacle)
            # check if the receptacle is closed. If so, open it.
```

```
                if 'closed' in observation:
                    observation = agent.open_receptacle(receptacle)
                # check if a bowl is in/on the receptacle.
                if 'pen' in observation:
                    break
            # expectation: I should be able to find a receptacle where a bowl is
            in/on it.
            assert 'pen' in observation, f'Error in [Step 2]: There is no bowl in/on
            {recep_to_check}. {agent.report()}'

        if start_from <= 3:
            print("[Step 3] take the bowl from the receptacle")
            # I need to get the identifier of the bowl so that I can take it. I can
            ask the assistant to do that.
            answer = ask(f'From the observation, get the identifier of an object.
            For example, On the cabinet 1, you see a cloth 2, and a toiletpaper 2.
            The identifier of cloth is 2. Now, {observation} The identifier of the
            pen? Only Output a single number without any other words. ')
            found_pen = f'pen {answer}'
            observation = agent.take(found_pen, receptacle)
            # expectation: I should be able to take the bowl from the receptacle.
            assert agent.holding == found_pen, f'Error in [Step 3]: I cannot take
            {found_pen} from the {receptacle}. {agent.report()}'

        if start_from <= 4:
            print("[Step 4] get a list of receptacles where a desklamp is likely to
            appear.")
            # I can ask the assistant to do that.
            answer = ask(f'Given a list of receptacles, please sort them in
            descending order based on the likelihood of finding a desklamp in each
            of them. The list of receptacles is: {agent.receptacles}. You should
            directly return a Python list.')
            recep_to_check = literal_eval(answer)
            # expectation: the returned recep_to_check should not be empty.
            assert recep_to_check, f'Error in [Step 4]: recep_to_check should not be
            empty. {agent.report()}'

        if start_from <= 5:
            print("[Step 5] go to each receptacle in the list until seeing a
            desklamp")
            for receptacle in recep_to_check:
                observation = agent.goto(receptacle)
                # check if the receptacle is closed. If so, open it.
                if 'closed' in observation:
                    observation = agent.open_receptacle(receptacle)
                # check if a desklamp is in/on the receptacle.
                if 'desklamp' in observation:
                    break
            # expectation: I should be able to find a receptacle where a desklamp is
            in/on it.
            assert 'desklamp' in observation, f'Error in [Step 5]: There is no
            desklamp in/on {recep_to_check}. {agent.report()}'

        if start_from <= 6:
            print("[Step 6] turn on desklamp")
            # There might be multiple desklamps in the environment, and I need to
            get the identifier of the desklamp. I can ask the assistant to do that.
            answer = ask(f'From the observation, get the identifier of an object.
            For example, On the cabinet 1, you see a cloth 2, and a toiletpaper 2.
            The identifier of cloth is 2. Now, {observation} The identifier of the
            desklamp? Only Output a single number without any other words.')
            found_desklamp = f'desklamp {answer}'
            # I can directly turn on the desklamp that I just found.
            observation = agent.turn_on(found_desklamp)
```

```
            # expectation: the desklamp should be turned on now.
            assert 'turn on' in observation, f'Error in [Step 6]: I cannot turn on
            {found_desklamp} in/on {receptacle}. {agent.report()}'
```

The expert sample for the task `Pick two`:

```
┌─ <sample_picktwo> Prompt ─┐
```

```python
# define environment and agent
receptacles = ['diningtable 1','drawer 2', 'drawer 1', 'sinkbasin 1', 'toilet
1', 'sidetable 2', 'sidetable 1', 'cabinet 1', 'countertop 1', 'microwave 1',
'fridge 1']
agent = Agent(receptacles)

# Your task is to: put two cellphone in cabinet / find two cellphone and put
them in cabinet
# here is a solution:
def solution(agent, start_from=1):
    if start_from <= 1:
        print("[Step 1] get a list of receptacles where a cellphone is likely to
        appear.")
        # I can ask the assistant to do that.
        answer = ask(f'Given a list of receptacles, please sort them in
        descending order based on the likelihood of finding a cellphone in each
        of them. The list of receptacles is: {agent.receptacles}. You should
        directly return a Python list.')
        recep_to_check = literal_eval(answer)
        # remove the destination from the list
        recep_to_check.remove('cabinet 1')
        # expectation: the returned recep_to_check should not be empty.
        assert recep_to_check, f'Error in [Step 1]: recep_to_check should not be
        empty. {agent.report()}'

    if start_from <= 2:
        print("[Step 2] go to each receptacle in the list until seeing a
        cellphone")
        for receptacle in recep_to_check:
            observation = agent.goto(receptacle)
            # check if the receptacle is closed. If so, open it.
            if 'closed' in observation:
                observation = agent.open_receptacle(receptacle)
            # check if a cellphone is in/on the receptacle.
            if 'cellphone' in observation:
                break
        # expectation: I should be able to find a receptacle where a cellphone
        is in/on it.
        assert 'cellphone' in observation, f'Error in [Step 2]: There is no
        cellphone in/on {recep_to_check}. {agent.report()}'

    if start_from <= 3:
        print("[Step 3] identify the first cellphone found and take it")
        # I need to get the identifier of the cellphone. I can ask the assistant
        to do that.
        answer = ask(f'From the observation, get the identifier of an object.
        For example, On the cabinet 1, you see a cloth 2, and a toiletpaper 2.
        The identifier of cloth is 2. Now, {observation}. The identifier of the
        cellphone? Only Output a single number without any other words. ')
        found_cellphone1 = f'cellphone {answer}'
        observation = agent.take(found_cellphone1, receptacle)
        # expectation: I should be able to take the cellphone from the
        receptacle.
        assert agent.holding == found_cellphone1, f'Error in [Step 3]: I cannot
        take {found_cellphone1} from the {receptacle}. {agent.report()}'

    if start_from <= 4:
```

```
        print("[Step 4] go to a cabinet and put the first cellphone found on it.
        ")
        # There are multiple countertops, and I only need to go to one of them.
        observation = agent.goto('cabinet 1')
        # check if the cabinet is closed. If so, open it.
        if 'closed' in observation:
            observation = agent.open_receptacle('cabinet 1')
        observation = agent.put(found_cellphone1, 'cabinet 1')
        # expectation: I should be able to put the cellphone1 on the countertop.
        assert f'You put the {found_cellphone1} in/on the cabinet 1.' in
        observation, f'Error in [Step 4]: I cannot put the {found_cellphone1} on
        the cabinet 1. {agent.report()}'

    if start_from <= 5:
        print("[Step 5] go to each of the remaining receptacle in the list until
        seeing a second cellphone")
        for receptacle in recep_to_check:
            observation = agent.goto(receptacle)
            # check if the receptacle is closed. If so, open it.
            if 'closed' in observation:
                observation = agent.open_receptacle(receptacle)
            # check if a cellphone is in/on the receptacle.
            if 'cellphone' in observation:
                break
        # expectation: I should be able to find a receptacle where a cellphone
        is in/on it.
        assert 'cellphone' in observation, f'Error in [Step 5]: There is no
        second cellphone in/on {recep_to_check}. {agent.report()}'

    if start_from <= 6:
        print("[Step 6] identify the second cellphone I just found and take it")
        # I need to get the identifier of the cellphone. I can ask the assistant
        to do that.
        answer = ask(f'From the observation, get the identifier of an object.
        For example, On the cabinet 1, you see a cloth 2, and a toiletpaper 2.
        The identifier of cloth is 2. Now, {observation}. The identifier of the
        cellphone? Only Output a single number without any other words. ')
        found_cellphone2 = f'cellphone {answer}'
        observation = agent.take(found_cellphone2, receptacle)
        # expectation: I should be able to take the cellphone from the
        receptacle.
        assert agent.holding == found_cellphone2, f'Error in [Step 6]: I cannot
        take {found_cellphone2} from the {receptacle}. {agent.report()}'

    if start_from <= 7:
        print("[Step 7] go to a cabinet and put the second cellphone found on
        it")
        observation = agent.goto('cabinet 1')
        observation = agent.put(found_cellphone2, 'cabinet 1')
        # expectation: I should be able to put the cellphone2 on the countertop.
        assert f'You put the {found_cellphone2} in/on the cabinet 1.' in
        observation, f'Error in [Step 7]: I cannot put the {found_cellphone2} on
        the cabinet 1. {agent.report()}'
```

**Code Check.** After plan generation, we employ the following prompt to verify and rectify any syntax errors. The placeholder <solution_func> is replaced by the generated solution function. The <code_check> prompt prompts the model to return two questions. If the response to Question 1 is Yes, the answer to Question 2 is adopted as the corrected solution function. Otherwise, the solution function is kept unchanged.

---

| <code_check> Prompt |

You are given a Python code snippet to define a function called solution.

[Code]

```
<solution_func>

Question 1: Are there any syntax errors present in the code? Answer Yes/No.
Question 2: Fix the syntax errors and output an error-free version of the code.
Only Output the revised code after [Revised code] without any other words.
```

**Out-of-Plan Refinement.** In the event of an assertion error, we use `<refinement>` to conduct the out-of-plan refinement. In this prompt, `<basic_info>` is replaced by the content of the `<basic_info>` prompt. The placeholder `<sample>` is substituted with an expert trajectory, while `<receptacle_list>` is replaced by the list of interactive receptacles provided by the task environment. `<task>` is replaced by the task description in natural language. Finally, `<error_msg>` is replaced by the assertion error message returned by the solution function. To adhere to the context length limit of the GPT-3/3.5 models, the previously generated solution function is not included in this prompt. Instead, we incorporate comprehensive information in the assertion error message, enabling the refiner to generate a revised plan based on these details.

---
`<refinement>` Prompt
---
```
<basic_info>

# Here is a example of successful solution for solving a similar task:
[Successful example]
receptacles = ['diningtable 1','drawer 2', 'drawer 1', 'sinkbasin 1', 'toilet
1', 'sidetable 2', 'sidetable 1', 'cabinet 1', 'countertop 1', 'microwave 1',
'fridge 1']
agent = Agent(receptacles)
<sample>

# Here is the actual task.
# define environment and agent
receptacles = <receptacle_list>
agent = Agent(receptacles)

# <task>
You have generated code of solution() to solve the task. However, you executed
the solution() function and get an error message:
<error_msg>

Let's think step by step. Referring to the successful case and the error
message, you should complete the solution function with the correct code.
def solution(agent, start_from=1):
```

**Determining** `start_from`. After formulating a revised plan, we utilize the following prompt to ascertain from which step the new solution function should commence. In this context, the `<previous_solution>` is replaced by the preceding solution function, while the `<revised_solution>` is replaced by the updated one. Subsequently, the argument `start_from=1` is substituted with the step number that this prompt yields.

---
`<start_from>` Prompt
---
```
Previously, you generated some code defining a solution function as in [Previous
solution]. The previous code is executed and outputs some error. Now you just
revised the code as in [Revised solution]. Determine from which step these two
version differs. You should only output the step number without saying any other
words.

[Previous solution]
<previous_solution>

[Revised solution]
<revised_solution>
```

## D.2 MiniWoB++

**Basic Information.** Similar to the ALFWorld tasks, the `<basic_info>` of MiniWoB++ defines the agent and admissible actions for AdaPlanner. Note that the actual definitions of action functions are not specified in the prompt. Instead, only a formalized definition with several examples is provided, such that the planner can acquire how to compose a plan based on these actions. As can be seen in the following part, this `<basic_info>` prompt is used in both `<initial_planning>` and `<refinement>` prompts.

```
                        ┌─────────────────────────┐
────────────────────────│  <basic_info> Prompt   │────────────────────────
                        └─────────────────────────┘
# Interact with the HTML webpage to finish the computer task. Here is some
Python code defining a computer task environment:

# In the environment, you can ask questions to an assistant by ask():
from large_language_model import ask_llm as ask
# for example: You want to solve a algebra problem x + 3 = 6. You can ask the
assistant to solve it directly for you.
answer = ask('Solve an algebra problem. You should directly output the value of
the unknown. For example, solve 'y + 1 = 2' -> 1. Now solve 'x + 3 = 6' ->')
# answer = '3'

# Agent class represents the current state of the HTML webpage and the actions
it can take.
class Agent:
    def __init__(self, initial_state):
        self.current_state = initial_state

    # Here are the admissible actions the agent can take:
    # Action: type a string into the input box
    # this function returns the string of the HTML code after taking the action
    # e.g., new_html_state = agent.type("hello")
    def type(self, characters: str) -> str:
        ...

    # Action: press a key on the keyboard, the input can be one of the following:
    # enter, space, arrow_left, arrow_right, arrow_up, arrow_down, backspace
    # this function returns the string of the HTML code after taking the action
    # e.g., new_html_state = agent.press_key("enter")
    def press_key(self, key: str) -> str:
        ...

    # Action: click a <select> element in a list with an XPath
    # this function returns the string of the HTML code after taking the action
    # e.g., new_html_state = agent.click_option("//select[@id='cars']/option[1]")
    def click_option(self, xpath_of_option: str) -> str:
        ...

    # Action: click an HTML element with its XPath
    # this function returns the string of the HTML code after taking the action
    # e.g., new_html_state = agent.click_xpath("//button[@id='button1']")
    def click_xpath(self, xpath_of_element: str) -> str:
        ...

    # Action: move the mouse cursor on an HTML element with an XPath
    # this function returns the string of the HTML code after taking the action
    # e.g., new_html_state = agent.move_mouse_on("//button[@id='button1']")
    def move_mouse_on(self, xpath_of_element: str) -> str:
        ...
```

**Initial Planning.** The `<initial_planning>` prompt is employed to generate the preliminary plan. In this context, `<basic_info>` is substituted by the content of the `<basic_info>` prompt. The `<sample>` is replaced with an expert trajectory (or discarded in the skill discovery stage). The `<initial_state>` is substituted by the initial HTML code of the task UI. Finally, `<task>` is substituted by the task description, expressed in natural language.

```
<basic_info>

<sample>

# Now complete the function solution() below to solve the task by composing the
agent's methods to interact with the environment.
# In the solution function, start with a commented "# General plan: ". For each
step you plan to take, mark with '[Step xx]', and write an assertion to check if
the step is successful.

# Here is the actual task.
# define environment and agent. The state is the list of HTML elements of the
webpage.
initial_state = \'''
<initial_state>
\'''
agent = Agent(initial_state)

# Task: <task>
# Here is the solution:
def solution(agent, start_from=1):
```

**Samples.** In MiniWoB++, we select nine tasks where feedback is available. Here are the expert samples that we gathered for these tasks.

The expert sample for the task email-inbox:

```
# Here are three examples of solutions.
# Task: Find the email by Brittani and reply to them with the text "Aliquet.
Sollicitudin nam lectus.".
def solution(agent, start_from=1):
# General plan: I should first click on the email by Brittani, then click on the
reply button, then type the text "Aliquet. Sollicitudin nam lectus." and finally
click on the send button.
    if start_from <= 1:
        print('[Step 1] click on the email by Brittani')
        agent.click_xpath("//div[@class='email-sender' and text()='Brittani']")
        state_after_interaction =
        agent.click_xpath("//span[@class='email-reply']")
        # the reply content should be displayed on page.
        assert 'reply-text' in state_after_interaction, 'I cannot do [Step 1]
        correctly. The reply button is not displayed on the page.'

    if start_from <= 2:
        print('[Step 2] type the text "Aliquet. Sollicitudin nam lectus."')
        agent.click_xpath("//textarea[@id='reply-text']")
        agent.type('Aliquet. Sollicitudin nam lectus.')
        state_after_interaction = agent.click_xpath("//*[@id='send-reply']")

# Task: Find the email by Blanca and forward that email to Agathe.
def solution(agent, start_from=1):
# General plan: I should first click on the email by Blanca, then click on the
forward button, then type "Agathe" and finally click on the send button.
    if start_from <= 1:
        print('[Step 1] click on the email by Blanca')
        agent.click_xpath("//div[@class='email-sender' and text()='Blanca']")
        state_after_interaction =
        agent.click_xpath("//span[@class='email-forward']")
        # the forward content should be displayed on page.
        assert 'forward-sender' in state_after_interaction, 'I cannot do [Step
        1] correctly. The forward button is not displayed on the page.'
```

```
        if start_from <= 2:
            print('[Step 2] type "Agathe"')
            agent.click_xpath("//input[@class='forward-sender']")
            agent.type('Agathe')
            state_after_interaction = agent.click_xpath("//*[@id='send-forward']")

# Task: Find the email by Salli and click the trash icon to delete it.
def solution(agent, start_from=1):
# General plan: I should first click on the email by Salli, then click on the
trash icon.
    if start_from <= 1:
        print('[Step 1] click on the email by Salli')
        agent.click_xpath("//div[@class='email-sender' and text()='Salli']")
        agent.click_xpath("//span[@class='trash']")
```

The expert sample for the task `email-inbox-forward-nl`:

```
# Here is an example of solution.
# task: Send Alice the email from Beth / navigate to the message from Beth and
send it to Alice.
def solution(agent, start_from=1):
# General plan: I should first click on the email from Beth, then click on the
"Forward" button, then type "Alice" in the "To" inputbox, finally click on the
"Send" button.
    if start_from <= 1:
        print('[Step 1] click on the email from Beth')
        agent.click_xpath('//*[@class="email-sender" and text()="Beth"]')
        state_after_interaction =
        agent.click_xpath('//span[@class="email-forward"]')
        # the "To" inputbox should be displayed on page.
        assert 'forward-sender' in state_after_interaction, f'I cannot do [Step
        1] correctly. The "To" inputbox is not displayed on the page. Current
        state: {state_after_interaction}'

    if start_from <= 2:
        print('[Step 2] type "Alice" in the "To" inputbox')
        agent.click_xpath('//input[@class="forward-sender"]')
        agent.type('Alice')
        state_after_interaction = agent.click_xpath('//span[@id="send-forward"]')
        # the email should be sent successfully.
        assert 'email-sender' in state_after_interaction, f'I cannot do [Step 2]
        correctly. The email is not sent successfully. Current state:
        {state_after_interaction}'
```

The expert sample for the task `email-inbox-forward-nl-turk`:

```
# Here is an example of solution.
# task: Send Alice the email from Beth / navigate to the message from Beth and
send it to Alice / I want to forward the email from Beth over to Alice
def solution(agent, start_from=1):
# General plan: I should first click on the email from Beth, then click on the
"Forward" button, then type "Alice" in the "To" inputbox, finally click on the
"Send" button.
    if start_from <= 1:
        print('[Step 1] click on the email from Beth')
        agent.click_xpath('//*[@class="email-sender" and text()="Beth"]')
        state_after_interaction =
        agent.click_xpath('//span[@class="email-forward"]')
        # the "To" inputbox should be displayed on page.
        assert 'forward-sender' in state_after_interaction, f'I cannot do [Step
        1] correctly. The "To" inputbox is not displayed on the page. Current
        state: {state_after_interaction}'
```

```
    if start_from <= 2:
        print('[Step 2] type "Alice" in the "To" inputbox')
        agent.click_xpath('//input[@class="forward-sender"]')
        agent.type('Alice')
        state_after_interaction = agent.click_xpath('//span[@id="send-forward"]')
        # the email should be sent successfully.
        assert 'email-sender' in state_after_interaction, f'I cannot do [Step 2]
        correctly. The email is not sent successfully. Current state:
        {state_after_interaction}'
```

The expert sample for the task email-inbox-nl-turk:

```
┌──────────────── <sample_email-inbox-nl-turk> Prompt ────────────────┐

# Here are three examples of solution.
# Task: "Aliquet. Sollicitudin nam lectus." is my reply to Brittani's most
recent email / Find the email by Brittani and reply to them with the text
"Aliquet. Sollicitudin nam lectus.".
def solution(agent, start_from=1):
# General plan: I should first click on the email by Brittani, then click on the
reply button, then type the text "Aliquet. Sollicitudin nam lectus." and finally
click on the send button.
    if start_from <= 1:
        print('[Step 1] click on the email by Brittani')
        agent.click_xpath("//div[@class='email-sender' and text()='Brittani']")
        state_after_interaction =
        agent.click_xpath("//span[@class='email-reply']")
        # the reply content should be displayed on page.
        assert 'reply-text' in state_after_interaction, 'I cannot do [Step 1]
        correctly. The reply button is not displayed on the page.'

    if start_from <= 2:
        print('[Step 2] type the text "Aliquet. Sollicitudin nam lectus."')
        agent.click_xpath("//textarea[@id='reply-text']")
        agent.type('Aliquet. Sollicitudin nam lectus.')
        state_after_interaction = agent.click_xpath("//*[@id='send-reply']")

# Task: Find the last email by Blanca and send it to Agathe / Find the email by
Blanca and forward that email to Agathe.
def solution(agent, start_from=1):
# General plan: I should first click on the email by Blanca, then click on the
forward button, then type "Agathe" and finally click on the send button.
    if start_from <= 1:
        print('[Step 1] click on the email by Blanca')
        agent.click_xpath("//div[@class='email-sender' and text()='Blanca']")
        state_after_interaction =
        agent.click_xpath("//span[@class='email-forward']")
        # the forward content should be displayed on page.
        assert 'forward-sender' in state_after_interaction, 'I cannot do [Step
        1] correctly. The forward button is not displayed on the page.'

    if start_from <= 2:
        print('[Step 2] type "Agathe"')
        agent.click_xpath("//input[@class='forward-sender']")
        agent.type('Agathe')
        state_after_interaction = agent.click_xpath("//*[@id='send-forward']")

# Task: Delete this email from Salli / Please find Salli's email in the inbox
and delete it.
def solution(agent, start_from=1):
# General plan: I should first click on the email by Salli, then click on the
trash icon.
    if start_from <= 1:
        print('[Step 1] click on the email by Salli')
```

```
                        agent.click_xpath("//div[@class='email-sender' and text()='Salli']")
                        agent.click_xpath("//span[@class='trash']")
```

The expert sample for the task `guess-number`:

<sample_guess-number> Prompt

```
# Here is an example of solution.
# Task: Guess the number between 0-9 and press Submit. Use the feedback below to
find the right number.
# Here is the solution:
def solution(agent, start_from = 1):
# General plan: Given a list of possible_numbers, I will try the number in the
middle of the list and get feedback from the html state.
# Now the given list of possible_numbers is [0, 1, 2, 3, 4, 5, 6, 7, 8, 9].
    if start_from <= 1:
        print('[Step 1] maintain a list of the possible numbers left and try the
        middle one')
        # before making a guess, I should store the html state for future
        comparison.
        state_before_interaction = agent.current_state
        # I will choose the number 5, which is in the middle of possible_numbers.
        guess_number = 5
        # click the input box, type the number I guessed and click submit.
        agent.click_xpath("//input[@id='tt']")
        agent.press_key("backspace")
        agent.type(str(guess_number))
        state_after_interaction = agent.click_xpath('//*[@id="subbtn"]')
        # after input and submit the guessed number, the html_state should be
        changed and contain the feedback. Otherwise this step is not successful.
        assert state_after_interaction != state_before_interaction, 'I did [Step
        1] but the html state did not change.'

    if start_from <= 2:
        print('[Step 2] get the feedback information from the new html state')
        # If the guess is successful, the keyword "higher" or "lower" should not
        be present in the observation. Otherwise I should use assert to jump out
        to pass the feedback.
        observation = ask(f'Answer a question based on the html code below:
        {state_after_interaction} Question: Which one is displayed? "The number
        is lower than" or "The number is higher than"? You must only output the
        displayed sentence without saying any other words.')
        assert "higher" not in observation, f'You tried the number
        {guess_number} in [Step 1], and the correct number is greater than this
        number. I need to revise solution function according to the new plan:
        Now the given list of possible_numbers is [6, 7, 8, 9].'
        assert "lower" not in observation, f'You tried the number {guess_number}
        in [Step 1], and the correct number is smaller than this number. I need
        to revise solution function according to the new plan: Now the given
        list of possible_numbers is [0, 1, 2, 3, 4].'
```

The expert sample for the task `login-user-popup`:

<sample_login-user-popup> Prompt

```
# Here is an example of solution.
# Task: Enter the username "kanesha" and the password "oT" into the text fields
and press login.
def solution(agent, start_from=1):
# General plan: I should first click on the username field, then type in the
username, then click on the password field, then type in the password, then
click on the login button.
    if start_from <= 1:
        print('[Step 1] Click on the username field')
        state_after_interaction = agent.click_xpath("//input[@id=\'username\']")
```

```
            # during interaction, some popups may appear. If so, I need to jump out
            to handle the popups.
            assert 'popup' not in state_after_interaction, f'After [Step 1], some
            popups appeared, you need to close the popup at the beginning of [Step
            1]. The current html is: {state_after_interaction} You need to add some
            code at the beginning of [Step 1] to cancel the popup before any other
            actions.'

    if start_from <= 2:
        print('[Step 2] Type in the username')
        state_after_interaction = agent.type('kanesha')
        # during interaction, some popups may appear. If so, I need to jump out
        to handle the popups.
        assert 'popup' not in state_after_interaction, f'After [Step 2], some
        popups appeared, you need to close the popup at the beginning of [Step
        2]. The current html is: {state_after_interaction} You need to add some
        code at the beginning of [Step 2] to cancel the popup before any other
        actions.'

    if start_from <= 3:
        print('[Step 3] Click on the password field')
        state_after_interaction = agent.click_xpath("//input[@id=\'password\']")
        # during interaction, some popups may appear. If so, I need to jump out
        to handle the popups.
        assert 'popup' not in state_after_interaction, f'After [Step 3], some
        popups appeared, you need to close the popup at the beginning of [Step
        3]. The current html is: {state_after_interaction} You need to add some
        code at the beginning of [Step 3] to cancel the popup before any other
        actions.'

    if start_from <= 4:
        print('[Step 4] Type in the password')
        state_after_interaction = agent.type('oT')
        # during interaction, some popups may appear. If so, I need to jump out
        to handle the popups.
        assert 'popup' not in state_after_interaction, f'After [Step 4], some
        popups appeared, you need to close the popup at the beginning of [Step
        4]. The current html is: {state_after_interaction} You need to add some
        code at the beginning of [Step 4] to cancel the popup before any other
        actions.'

    if start_from <= 5:
        print('[Step 5] Click on the login button')
        state_after_interaction = agent.click_xpath("//button[@id='subbtn']")
        # during interaction, some popups may appear. If so, I need to jump out
        to handle the popups.
        assert 'popup' not in state_after_interaction, f'After [Step 5], some
        popups appeared, you need to close the popup at the beginning of [Step
        5]. The current html is: {state_after_interaction} You need to add some
        code at the beginning of [Step 5] to cancel the popup before any other
        actions.'
```

The expert sample for the task `search-engine`:

```
________________________ <sample_search-engine> Prompt ________________________
# Here is an example of solution.
# Task: Use the textbox to enter "Alice" and press "Search", then find and click
the 7th search result.
def solution(agent, start_from=1):
# General plan: I should first click on the inputbox, then type "Alice", then
click on the "Search" button, finally look through pages and click on the 7th
result.
    if start_from <= 1:
        print('[Step 1] click on the inputbox and type "Alice"')
```

```
        agent.click_xpath('//*[@id="search-text"]')
        agent.type('Alice')
        state_after_interaction = agent.click_xpath('//*[@id="search"]')
        # the search content should be displayed on page.
        assert 'search-desc' in state_after_interaction, 'I cannot do [Step 1]
        correctly. The search content is not displayed on the page.'

    if start_from <= 2:
        print('[Step 2] calculate the page number of the 7th result and click on
        the page')
        # I should count the number of results on each page, iteratively turn to
        next page until seeing the 7th result.
        # I can use the following code to count the number of results on each
        page.
        num_results_displayed_per_page =
        state_after_interaction.count('search-desc')
        num_page = (7 - 1) // num_results_displayed_per_page
        state_after_interaction =
        agent.click_xpath(f'//*[@id="pagination"]/li[{3+num_page}]/a')
        # I should click on the 7th result.
        num_to_click = 7 - num_results_displayed_per_page * num_page
        state_after_interaction =
        agent.click_xpath(f'//*[@id="page-content"]/div[{num_to_click}]/a')
```

The expert sample for the task `terminal`:

─────────────── <sample_terminal> Prompt ───────────────

```
# Here is an example of solution.
# Task: Use the terminal below to delete a file ending with the extension .gif
def solution(agent, start_from=1):
# General plan: I should first type "ls" to list all files in the terminal, then
identify the filename ending with ".gif" and type "rm [filename].gif" to delete
the identified file.
    if start_from <= 1:
        print('[Step 1] type "ls" to list all files in the terminal')
        agent.type('ls')
        state_after_interaction = agent.press_key('enter')
        # the file list should be displayed on terminal.
        assert 'gif' in state_after_interaction, f'I cannot do [Step 1]
        correctly. The file list is not displayed on the terminal. Current
        state: {state_after_interaction}'

    if start_from <= 2:
        print('[Step 2] identify the filename ending with ".gif" and type "rm
        [filename].gif" to delete the identified file')
        # I should identify the filename ending with ".gif". I can ask assistant
        to do that.
        filename = ask(f'You are given some html code as follows:
        {state_after_interaction} What is the file ending with the extension
        .gif? You must directly output the full file name, including the
        extension.')
        agent.type(f'rm {filename}')
        state_after_interaction = agent.press_key('enter')
        assert 'not found' not in state_after_interaction, f'I cannot do [Step
        2] correctly. The file ending with the extension .gif is not deleted.
        Current state: {state_after_interaction}'
```

The expert sample for the task `tic-tac-toe`:

─────────────── <sample_tic-tac-toe> Prompt ───────────────

```
# Here is an example of solution.
# Task: Playing as 'X', win a game of tic-tac-toe.
def solution(agent, start_from=1):
```

```
# The board layout and corresponding html xpath: top-left("//*[@id='ttt-0']"),
top-center("//*[@id='ttt-1']"), top-right("//*[@id='ttt-2']"),
middle-left("//*[@id='ttt-3']"), middle-center("//*[@id='ttt-4']"),
middle-right("//*[@id='ttt-5']"), bottom-left("//*[@id='ttt-6']"),
bottom-center("//*[@id='ttt-7']"), bottom-right("//*[@id='ttt-8']").  Note that
"mark-o" indicates the 'O' placed on board, "mark-x" indicates the 'X' placed on
board.
# General plan: Currently, no grid is occupied. The plan is 1) put an 'X' in the
middle-center("//*[@id='ttt-4']"), 2) put an 'X' in the
top-left("//*[@id='ttt-0']"), 3) put an 'X' in the
bottom-right("//*[@id='ttt-8']").
    place_to_put_X = ['4', '0', '8']
    for idx, place_id in enumerate(place_to_put_X):
        print(f'[Step {idx}] put an X in {place_id}')
        # before interaction, I need to store the current state so that I can
        compare it with the state after interaction.
        state_before_interaction = agent.current_state
        state_after_interaction = agent.click_xpath(f"//*[@id='ttt-{place_id}']")
        # if the current state does not change after interaction, that means I
        cannot put an 'X' in the desired location, and that location is already
        occupied and the plan will not work.
        assert state_before_interaction != state_after_interaction, f'''I cannot
        do [Step {idx}] put an X in the "//*[@id='ttt-{place_id}']", because it
        is occupied. I need to revise solution function according to the new
        plan. ''' + ask(f'''Playing as 'X', win a game of tic-tac-toe. The board
        layout and corresponding html xpath: top-left("//*[@id='ttt-0']"),
        top-center("//*[@id='ttt-1']"), top-right("//*[@id='ttt-2']"),
        middle-left("//*[@id='ttt-3']"), middle-center("//*[@id='ttt-4']"),
        middle-right("//*[@id='ttt-5']"), bottom-left("//*[@id='ttt-6']"),
        bottom-center("//*[@id='ttt-7']"), bottom-right("//*[@id='ttt-8']").Note
        that "mark-o" indicates the 'O' placed on board, "mark-x" indicates the
        'X' placed on board. The game in progress is represented in html code:
        {agent.current_state} Report current board situation and generate a plan
        that the 'X' player should follow to continue this game. Use the format
        like "Currently, 'X' has been placed at <position>("//*[@id='ttt-x']")
        and 'O' has been placed at <position>("//*[@id='ttt-x']"). Therefore,
        the plan is to: 1) put an 'X' in the <position>("//*[@id='ttt-x']") 2)
        put an 'X' in the ..."''')
```

**Code Check.** We use the same `<code_check>` prompt for MiniWoB++ tasks as ALFWorld.

**Out-of-Plan Refinement.** In the event of an assertion error, we use `<refinement>` to conduct the out-of-plan refinement. In this prompt, `<basic_info>` is replaced by the content of the `<basic_info>` prompt. The placeholder `<solution_func>` is replaced by the generated solution function, while `<task>` is replaced by the task description in natural language. Finally, `<feedback>` is replaced by the assertion error message returned by the solution function. Note that the `<refinement>` prompt used here differs slightly from the one used for ALFWorld. In this context, we include the previously generated solution function in the prompt, enabling the refiner to adjust the solution based on the error messages accordingly.

---
`<refinement>` Prompt
---

```
<basic_info>

# Task: <task>
You have generated code of solution() to solve the task:
<solution_func>

However, you executed the solution() function and get an error message:
[Error message]
<feedback>

Let's think step by step. You must output the revised solution function based on
the error message. You must only complete the revised solution function without
any other words.
```

```
# In the solution function, start with a commented "# General plan: ". For each
step you plan to take, mark with '[Step xx]', and write an assertion to check if
the step is successful.
def solution(agent, start_from=1):
```

**Determining** `start_from`. We use the same `<start_from>` prompt for MiniWoB++ tasks as
ALFWorld.

## E  Hallucination with GPT-3.5 Model (`gpt-3.5-turbo`)

We present the trajectories of three tested methods (ReAct, Reflexion, and AdaPlanner) for two tasks.
Here, each model utilizes `gpt-3.5-turbo` as the backend.

### E.1  Case 1: `examine the book with the desklamp`.

ReAct's trajectory for this case is detailed below. Despite the presence of two expert demonstrations,
ReAct encounters significant difficulties, as it extensively hallucinates inadmissible actions or sce-
narios. For example, `search for a book titled "The Great Gatsby"` and `ask household
members if they know where the book is` are not feasible within the given environment.

ReAct Trajectory of Case 1 (Hallucinated actions are marked in pink.)

```
You are in the middle of a room. Looking quickly around you, you see a bed 1, a
    desk 2, a desk 1, a drawer 6, a drawer 5, a drawer 4, a drawer 3, a drawer 2,
    a drawer 1, a garbagecan 1, a laundryhamper 1, a safe 1, a shelf 6, a shelf 5,
    a shelf 4, a shelf 3, a shelf 2, and a shelf 1.
Your task is to: examine the book with the desklamp.
> think: To solve the task, I need to find and take a book, then find and use a
    desklamp. First, I need to find a book. A book is more likely to appear in
    desk (1-2), shelf (1-6), drawer (1-6). I can check one by one, starting with
    desk 1.
OK.
> go to desk 1
On the desk 1, you see a alarmclock 1, a bowl 2, a cd 1, a cellphone 3, a desklamp
    1, a mug 1, a pen 1, and a pencil 1.
> take book from desk 1
Nothing happens.
> think: I need to be more specific about which book I am looking for. Let me try
    looking for a specific title or author.
OK.
```
> `search for a book titled "The Great Gatsby"`
```
Nothing happens.
> think: Let me try looking for a book by a specific author. How about "To Kill a
    Mockingbird" by Harper Lee?
OK.
```
> `search for a book by Harper Lee`
```
Nothing happens.
> think: Maybe I should try looking in a different location. Let me check shelf 1.
OK.
> go to shelf 1
On the shelf 1, you see a bowl 1.
> go to shelf 2
On the shelf 2, you see a creditcard 1, and a pencil 2.
> go to shelf 3
On the shelf 3, you see a alarmclock 1, a bowl 2, and a desklamp 1.
> take book from shelf 3
Nothing happens.
> think: Let me try looking in a different location. How about drawer 1?
OK.
> go to drawer 1
The drawer 1 is closed.
> open drawer 1
You open the drawer 1. The drawer 1 is open. In it, you see nothing.
> go to drawer 2
On the drawer 2, you see a cellphone 2.
> go to drawer 3
```

```
Nothing happens.
> go to drawer 4
The drawer 4 is closed.
> open drawer 4
You open the drawer 4. The drawer 4 is open. In it, you see nothing.
> go to drawer 5
On the drawer 5, you see nothing.
> go to drawer 6
The drawer 6 is closed.
> open drawer 6
You open the drawer 6. The drawer 6 is open. In it, you see nothing.
> think: Let me try looking in a different location. How about desk 2?
OK.
> go to desk 2
On the desk 2, you see a alarmclock 3, a bowl 3, a cd 3, a mug 2, and a pencil 3.
> take book from desk 2
Nothing happens.
> think: Let me try looking in a different location. How about shelf 4?
OK.
> go to shelf 4
On the shelf 4, you see a alarmclock 2.
> think: Let me try looking in a different location. How about shelf 5?
OK.
> go to shelf 5
On the shelf 5, you see nothing.
> think: Let me try looking in a different location. How about shelf 6?
OK.
> go to shelf 6
On the shelf 6, you see nothing.
> think: It seems like the book I am looking for is not in any of the locations I
    have checked so far. Let me ask the household members if they know where it is.
OK.
```

> ask household members if they know where the book is

```
Nothing happens.
> think: Maybe I should try asking a specific household member. How about asking
    John?
OK.
```

> ask John if he knows where the book is

```
Nothing happens.
> think: It seems like I am unable to find the book. I may need to ask for more
    information or try a different approach to the task.
OK.
```

> ask the household members if they have any suggestions on where to look for book

```
Nothing happens.
> think: It looks like I may need to give up on this task for now and come back to
    it later with more information or a different approach.
OK.
```

> end task

```
Nothing happens.
```

> exit room

```
Nothing happens.
```

> end interaction

```
Nothing happens.
```

> leave the household

```
Nothing happens.
```

> end interaction and try again later

```
Nothing happens.
```

> end interaction and come back later with a new approach

```
Nothing happens.
```

> end interaction and move on to a different task or activity

```
Nothing happens.
```

> end interaction and leave the room/household/location

```
Nothing happens.
```

> end interaction and try again later with a different approach or more information

```
Nothing happens.
```

> end interaction and seek help from someone outside of the household

```
Nothing happens.
> end interaction and try to find book in a different room or location in the house
Nothing happens.
```

The trajectory of Reflexion, after two rounds of iterative trials and errors, is detailed below. Despite multiple closed-loop corrections, Reflexion fails to formulate a feasible plan. Instead, it persists in generating unfeasible actions in its reflection part. For instance, it plans to ask the household members for help, or to look for a ladder or step stool outside of the room. These actions fall outside the scope of admissible actions in the ALFWorld environment.

Reflexion Trajectory of Case 1 (Hallucinated actions are marked in pink.)

```
Your memory for the task below:
Trial 0:
In this environment, my plan was to find a book then find and use a desklamp.
    However, I spent too much time checking every possible location without
    thoroughly examining each one. I should have focused on the most likely
    locations for a book (desk, shelf, drawer) and thoroughly searched each one
    before moving on to the next. Additionally, I should have asked the household
    members for help earlier in the process. In the next trial, I will prioritize
    searching the most likely locations and ask for help if I am unable to find
    the book after a thorough search.
Trial 1:
In the next trial, I will prioritize searching the most likely locations for a book
    (desk, shelf, drawer) and thoroughly examine each one before moving on to the
    next. If I am unable to find the book after a thorough search, I will ask the
    household members for help. If the book is out of reach, I will look for a
    ladder or step stool outside of the room that I can bring in to reach it.
Here is the task:
You are in the middle of a room. Looking quickly around you, you see a bed 1, a
    desk 2, a desk 1, a drawer 6, a drawer 5, a drawer 4, a drawer 3, a drawer 2,
    a drawer 1, a garbagecan 1, a laundryhamper 1, a safe 1, a shelf 6, a shelf 5,
    a shelf 4, a shelf 3, a shelf 2, and a shelf 1.
Your task is to: examine the book with the desklamp.
> think: To solve the task, I need to find and take a book, then find and use a
    desklamp. First, I need to find a book. A book is more likely to appear in
    desk (1-2), shelf (1-6), drawer (1-6). I can check one by one, starting with
    desk 1.
OK.
> go to desk 1
On the desk 1, you see a alarmclock 1, a bowl 2, a cd 1, a cellphone 3, a desklamp
    1, a mug 1, a pen 1, and a pencil 1.
> examine books on desk 1
Nothing happens.
> examine shelf 1
Nothing happens.
> go to shelf 2
On the shelf 2, you see a creditcard 1, and a pencil 2.
> go to shelf 3
On the shelf 3, you see a alarmclock 1, a bowl 2, and a desklamp 1.
> take book from shelf 3
Nothing happens.
> examine book on shelf 3
Nothing happens.
> ask household member for help
Nothing happens.
> think: I need to keep searching for the book. It could be in desk 2, shelf 4-6,
    or drawer 1-6. I will check drawer 1 next.
OK.
> go to drawer 1
The drawer 1 is closed.
> open drawer 1
You open the drawer 1. The drawer 1 is open. In it, you see nothing.
> go to drawer 2
On the drawer 2, you see a cellphone 2.
> take book from drawer 2
Nothing happens.
> examine book in drawer 2
```

```
Nothing happens.
> think: The book is not in drawer 2. I will check drawer 3 next.
OK.
> go to drawer 3
Nothing happens.
> examine books in drawer 3
Nothing happens.
> think: The book is not in drawer 3. I will check drawer 4 next.
OK.
> go to drawer 4
The drawer 4 is closed.
> open drawer 4
You open the drawer 4. The drawer 4 is open. In it, you see nothing.
> think: The book is not in drawer 4. I will check drawer 5 next.
OK.
> go to drawer 5
On the drawer 5, you see nothing.
> think: The book is not in drawer 5. I will check drawer 6 next.
OK.
> go to drawer 6
The drawer 6 is closed.
> open drawer 6
You open the drawer 6. The drawer 6 is open. In it, you see nothing.
> think: The book is not in drawer 6. I will check desk 2 next.
OK.
> go to desk 2
On the desk 2, you see a alarmclock 3, a bowl 3, a cd 3, a mug 2, and a pencil 3.
> think: The book is not on desk 2. I will check shelf 4 next.
OK.
> go to shelf 4
On the shelf 4, you see a alarmclock 2.
> think: The book is not on shelf 4. I will check shelf 5 next.
OK.
> go to shelf 5
On the shelf 5, you see nothing.
> think: The book is not on shelf 5. I will check shelf 6 next.
OK.
> go to shelf 6
On the shelf 6, you see nothing.
> think: The book is not on shelf 6. I have searched all the most likely locations
    for a book and have not found it. I will ask the household members for help.
OK.
> ask household member for help
Nothing happens.
> think: The household member did not respond. I will look for a ladder or step
    stool outside of the room that I can bring in to reach higher shelves or areas.
OK.
> go outside of the room
Nothing happens.
> look for a ladder or step stool
Nothing happens.
> think: I was unable to find a ladder or step stool. I will continue searching the
    room for the book and ask the household members for help again if necessary.
OK.
> go to desk 1
On the desk 1, you see a alarmclock 1, a bowl 2, a cd 1, a cellphone 3, a desklamp
    1, a mug 1, a pen 1, and a pencil 1.
> examine books on desk 1
Nothing happens.
> think: I have searched the room thoroughly and have not found the book. I will
    ask the household members for help again.
OK.
> ask household member for help
Nothing happens.
> think: The household member is not responding. I will try to think of any other
    possible locations for the book and search those areas.
OK.
> think: The book could be in the laundry hamper or safe. I will check those areas
    next.
OK.
```

```
| > go to laundry hamper 1                                                     |
| Nothing happens.                                                             |
| > examine laundry hamper 1                                                   |
| Nothing happens.                                                             |
| > go to safe 1                                                               |
| The safe 1 is closed.                                                        |
| > try to open safe 1                                                         |
| Nothing happens.                                                             |
```

The trajectory of AdaPlanner is detailed below. For better readability, we only present the major sub-goals generated by the planner. Evidently, AdaPlanner successfully completes this task without the need for any revisions. This indicates that the code interface can effectively mitigate the hallucination in `gpt-3.5-turbo`.

AdaPlanner Trajectory of Case 1

```
You are in the middle of a room. Looking quickly around you, you see a bed 1, a
    desk 2, a desk 1, a drawer 6, a drawer 5, a drawer 4, a drawer 3, a drawer 2,
    a drawer 1, a garbagecan 1, a laundryhamper 1, a safe 1, a shelf 6, a shelf 5,
    a shelf 4, a shelf 3, a shelf 2, and a shelf 1.
Your task is to: examine the book with the desklamp.
[Step 1] get a list of receptacles where a book is likely to appear.
[Step 2] go to each receptacle in the list until seeing a book
> go to shelf 6
On the shelf 6, you see nothing.
> go to shelf 5
On the shelf 5, you see nothing.
> go to shelf 4
On the shelf 4, you see a mug 2, and a pencil 1.
> go to shelf 3
On the shelf 3, you see a creditcard 1, a desklamp 1, and a mug 1.
> go to shelf 2
On the shelf 2, you see a pen 1.
> go to shelf 1
On the shelf 1, you see a bowl 1.
> go to desk 2
On the desk 2, you see a cellphone 2, a creditcard 2, a mug 3, and a pencil 2.
> go to desk 1
On the desk 1, you see a alarmclock 1, a creditcard 1, a desklamp 1, a laptop 3,
    and a mug 1.
> go to bed 1
On the bed 1, you see a book 1, a cellphone 1, a laptop 2, a laptop 1, a pillow 2,
    and a pillow 1.
[Step 3] take the book from the receptacle
> take book 1 from bed 1
You pick up the book 1 from the bed 1.
[Step 4] get a list of receptacles where a desklamp is likely to appear.
[Step 5] go to each receptacle in the list until seeing a desklamp
> go to desk 1
On the desk 1, you see a alarmclock 1, a creditcard 1, a desklamp 1, a laptop 3,
    and a mug 1.
[Step 6] turn on desklamp
> use desklamp 1
You turn on the desklamp 1.
```

**E.2 Case 2:** `put some watch on safe.`

In the second trajectory, ReAct also hallucinates infeasible actions (e.g., `look for a manual or instructions` and `ask someone in the household`), ultimately leading to task failure.

ReAct Trajectory of Case 2 (Hallucinated actions are marked in pink.)

```
You are in the middle of a room. Looking quickly around you, you see a armchair 1,
    a cabinet 4, a cabinet 3, a cabinet 2, a cabinet 1, a drawer 5, a drawer 4, a
    drawer 3, a drawer 2, a drawer 1, a dresser 1, a garbagecan 1, a safe 1, a
    shelf 12, a shelf 11, a shelf 10, a shelf 9, a shelf 8, a shelf 7, a shelf 6,
    a shelf 5, a shelf 4, a shelf 3, a shelf 2, a shelf 1, a sidetable 1, and a
    sofa 1.
```

```
Your task is to: put some watch on safe.
> think: To solve the task, I need to find and take a watch, then put it on the
    safe.
OK.
> think: First I need to find a watch. A watch is more likely to appear in a drawer
    (1-5), a shelf (1-12), a cabinet (1-4), a dresser (1), a sidetable (1), or a
    sofa (1). I can check one by one, starting with drawer 1.
OK.
> go to drawer 1
The drawer 1 is closed.
> open drawer 1
You open the drawer 1. The drawer 1 is open. In it, you see nothing.
> go to drawer 2
The drawer 2 is closed.
> open drawer 2
You open the drawer 2. The drawer 2 is open. In it, you see nothing.
> go to drawer 3
The drawer 3 is closed.
> open drawer 3
You open the drawer 3. The drawer 3 is open. In it, you see a creditcard 2.
> go to drawer 4
The drawer 4 is closed.
> open drawer 4
You open the drawer 4. The drawer 4 is open. In it, you see nothing.
> go to drawer 5
The drawer 5 is closed.
> open drawer 5
You open the drawer 5. The drawer 5 is open. In it, you see nothing.
> go to shelf 1
On the shelf 1, you see nothing.
> go to shelf 2
On the shelf 2, you see nothing.
> go to shelf 3
Nothing happens.
> go to shelf 4
On the shelf 4, you see nothing.
> go to shelf 5
On the shelf 5, you see nothing.
> go to shelf 6
On the shelf 6, you see nothing.
> go to shelf 7
On the shelf 7, you see a newspaper 3.
> go to shelf 8
On the shelf 8, you see nothing.
> go to shelf 9
On the shelf 9, you see nothing.
> go to shelf 10
On the shelf 10, you see nothing.
> go to shelf 11
On the shelf 11, you see nothing.
> go to shelf 12
On the shelf 12, you see nothing.
> go to cabinet 1
The cabinet 1 is closed.
> open cabinet 1
You open the cabinet 1. The cabinet 1 is open. In it, you see a vase 1.
> go to cabinet 2
Nothing happens.
> go to cabinet 3
The cabinet 3 is closed.
> open cabinet 3
You open the cabinet 3. The cabinet 3 is open. In it, you see a tissuebox 2.
> go to cabinet 4
Nothing happens.
> go to dresser 1
On the dresser 1, you see a laptop 1, a remotecontrol 1, a tissuebox 1, a vase 3, a
    vase 2, and a watch 1.
> take watch 1 from dresser 1
You pick up the watch 1 from the dresser 1.
> think: Now I have a watch (1). Next, I need to put it on the safe.
OK.
> go to safe 1
```

```
The safe 1 is closed.
> open safe 1
You open the safe 1. The safe 1 is open. In it, you see a cellphone 3, and a
    cellphone 2.
> put watch 1 on safe 1
Nothing happens.
> think: I need to be more specific about where to put the watch. Maybe on top of
    the cellphone 3?
OK.
> put watch 1 on cellphone 3 in safe 1
Nothing happens.
> think: Maybe I need to place the watch inside the safe instead of on top of
    something inside it.
OK.
> put watch 1 in safe 1
Nothing happens.
> think: Maybe I need to enter a code to properly open the safe and put the watch
    inside.
OK.
> look for a code or key to open the safe
Nothing happens.
> think: Maybe I need to ask someone in the household for the code or key to open
    the safe.
OK.
> ask someone in the household for the code or key to open the safe
Nothing happens.
> think: Maybe I need to search for a manual or instructions on how to open the
    safe.
OK.
> look for a manual or instructions on how to open the safe
Nothing happens.
> think: Maybe I need to try a combination of numbers to open the safe.
OK.
> try the combination 1234 on the safe
Nothing happens.
> think: Maybe I need to try a different combination of numbers to open the safe.
OK.
```

Similarly, Reflexion experiences issues with hallucination during this task, such as `seek additional resources` or `ask for help`. This contributes to the eventual failure of the task.

Reflexion Trajectory of Case 2 (Hallucinated actions are marked in pink.)

```
Your memory for the task below:
Trial 0:
In this environment, my plan was to find a watch and put it on the safe. However, I
    did not consider that the safe may require a code or key to lock it. I should
    have looked for a code or lock mechanism on the safe or a key to the safe
    before attempting to put the watch inside. When I could not find a way to
    secure the watch inside the safe, I should have looked for another container
    or safe in the room. In the next trial, I will first look for a code or lock
    mechanism on the safe or a key to the safe. If I cannot find one, I will look
    for another container or safe in the room.
Trial 1:
To solve the task, I need to find a watch and a way to secure it inside the safe.
    First, I will look for a watch in drawers (1-5), cabinets (1-4), dressers (1),
    shelves (1-12), sidetables (1), and sofas (1). Once I find a watch, I will
    examine the safe for any clues or look for a key in other locations. If I
    cannot find a way to secure the watch inside the safe, I will look for another
    container or safe in the room. If I still cannot find a way to secure the
    watch, I will seek additional resources or ask for help.
Here is the task:
You are in the middle of a room. Looking quickly around you, you see a armchair 1,
    a cabinet 4, a cabinet 3, a cabinet 2, a cabinet 1, a drawer 5, a drawer 4, a
    drawer 3, a drawer 2, a drawer 1, a dresser 1, a garbagecan 1, a safe 1, a
    shelf 12, a shelf 11, a shelf 10, a shelf 9, a shelf 8, a shelf 7, a shelf 6,
    a shelf 5, a shelf 4, a shelf 3, a shelf 2, a shelf 1, a sidetable 1, and a
    sofa 1.
Your task is to: put some watch on safe.
```

```
> think: To solve the task, I need to find a watch and a way to secure it inside
      the safe. First, I will look for a watch in drawers (1-5), cabinets (1-4),
      dressers (1), shelves (1-12), sidetables (1), and sofas (1).
OK.
> go to drawer 1
The drawer 1 is closed.
> open drawer 1
You open the drawer 1. The drawer 1 is open. In it, you see nothing.
> go to drawer 2
The drawer 2 is closed.
> open drawer 2
You open the drawer 2. The drawer 2 is open. In it, you see nothing.
> go to drawer 3
The drawer 3 is closed.
> open drawer 3
You open the drawer 3. The drawer 3 is open. In it, you see a creditcard 2.
> go to drawer 4
The drawer 4 is closed.
> open drawer 4
You open the drawer 4. The drawer 4 is open. In it, you see nothing.
> go to drawer 5
The drawer 5 is closed.
> open drawer 5
You open the drawer 5. The drawer 5 is open. In it, you see nothing.
> go to cabinet 1
The cabinet 1 is closed.
> open cabinet 1
You open the cabinet 1. The cabinet 1 is open. In it, you see a vase 1.
> go to cabinet 2
Nothing happens.
> think: The cabinet 2 may be locked or empty. I should try another location.
OK.
> go to cabinet 3
The cabinet 3 is closed.
> open cabinet 3
You open the cabinet 3. The cabinet 3 is open. In it, you see a tissuebox 2.
> go to cabinet 4
Nothing happens.
> think: The cabinet 4 may be locked or empty. I should try another location.
OK.
> go to dresser 1
On the dresser 1, you see a laptop 1, a remotecontrol 1, a tissuebox 1, a vase 3, a
      vase 2, and a watch 1.
> think: Now I find a watch (1). Next, I need to examine the safe for any clues or
      look for a key in other locations.
OK.
> go to safe 1
The safe 1 is closed.
> think: Before attempting to put the watch inside the safe, I should look for a
      code or lock mechanism on the safe or a key to the safe.
OK.
> examine safe 1
The safe 1 is closed.
> look for key or code on safe 1
Nothing happens.
> think: I cannot find a key or code on the safe. I should look for another
      container or safe in the room.
OK.
> look for another container or safe in the room
Nothing happens.
> think: I still cannot find another container or safe in the room. I should seek
      additional resources or ask for help.
OK.
> seek additional resources or ask for help
Nothing happens.
> think: I may need to think of a different solution to the task. Perhaps I can
      place the watch near the safe instead of inside it.
OK.
> put watch 1 near safe 1
Nothing happens.
```

```
> think: The task is incomplete. I need to put the watch on the safe. I should try
    to find a way to secure the watch on top of the safe or find a container to
    put the watch in that can be placed on top of the safe.
OK.
> look for a container to put the watch in that can be placed on top of the safe
Nothing happens.
> think: I should try to find a way to secure the watch on top of the safe.
OK.
> look for a way to secure the watch on top of the safe
Nothing happens.
> think: I may need to think of a different solution to the task. Perhaps I can
    find a way to attach the watch to the safe using tape or string.
OK.
> look for tape or string
Nothing happens.
> think: I cannot find tape or string. I should seek additional resources or ask
    for help.
OK.
> seek additional resources or ask for help
Nothing happens.
> think: The task is incomplete. I need to put the watch on the safe. I should try
    to find a way to secure the watch on top of the safe or find a container to
    put the watch in that can be placed on top of the safe. Since I cannot find
    any resources or help, I will have to abandon the task for now and come back
    to it later with new ideas or resources.
OK.
> end task
Nothing happens.
> exit room
Nothing happens.
> think: I cannot exit the room until the task is complete or I have abandoned it.
    I will continue to look for a solution to the task.
OK.
> continue searching for a solution to the task
Nothing happens.
> think: I have exhausted all possible options and resources in the room. I should
    take a break and come back to the task later with a fresh perspective or seek
    help from someone else.
OK.
> end task
Nothing happens.
> exit room
Nothing happens.
> think: I cannot exit the room until the task is complete or I have abandoned it.
    I will take a break and come back to the task later with a fresh perspective
    or seek help from someone else.
OK.
> end task
Nothing happens.
> exit room
Nothing happens.
```

Initially, AdaPlanner also experiences hallucination, presuming that the `watch` is located in the `safe`. This incorrect assumption triggers an assertion error. AdaPlanner then revises the plan to better adapt to the actual environment. Following this refinement, the task is successfully completed.

AdaPlanner Trajectory of Case 2 (Hallucinated actions are marked in pink.)

```
You are in the middle of a room. Looking quickly around you, you see a armchair 1,
    a cabinet 4, a cabinet 3, a cabinet 2, a cabinet 1, a drawer 5, a drawer 4, a
    drawer 3, a drawer 2, a drawer 1, a dresser 1, a garbagecan 1, a safe 1, a
    shelf 12, a shelf 11, a shelf 10, a shelf 9, a shelf 8, a shelf 7, a shelf 6,
    a shelf 5, a shelf 4, a shelf 3, a shelf 2, a shelf 1, a sidetable 1, and a
    sofa 1.
Your task is to: put some watch on safe.
[Step 1] get a list of receptacles where the safe is likely to appear
```

```
[Step 2] go to each receptacle in the list until seeing a safe
> go to safe 1
The safe 1 is closed.
> open safe 1
You open the safe 1. The safe 1 is open. In it, you see a cellphone 3, and a
    cellphone 2.
[Step 3] identify the watch I just found and take it
> take watch 1 from safe 1
Nothing happens.
[Step 4] go to the safe and put the watch in it
> go to safe 1
Nothing happens.
> put watch 1 in/on safe 1
Nothing happens.

AssertionError: Error in [Step 4]: I cannot put the watch 1 in the safe 1. The last
    three interactions before error were:
> take watch 1 from safe 1
Nothing happens.
> go to safe 1
Nothing happens.
> put watch 1 in/on safe 1
Nothing happens.
I am at safe 1 and holding None.

Refinement is done with start_from = 1.
[Step 1] get a list of receptacles where the watch is likely to appear
[Step 2] go to each receptacle in the list until seeing a watch
> go to safe 1
Nothing happens.
> go to cabinet 1
The cabinet 1 is closed.
> open cabinet 1
You open the cabinet 1. The cabinet 1 is open. In it, you see a vase 1.
> go to cabinet 2
Nothing happens.
> go to cabinet 3
The cabinet 3 is closed.
> open cabinet 3
You open the cabinet 3. The cabinet 3 is open. In it, you see a tissuebox 2.
> go to cabinet 4
Nothing happens.
> go to dresser 1
On the dresser 1, you see a laptop 1, a remotecontrol 1, a tissuebox 1, a vase 3, a
    vase 2, and a watch 1.
[Step 3] identify the watch I just found and take it
> take watch 1 from dresser 1
You pick up the watch 1 from the dresser 1.
[Step 4] go to the safe and put the watch in it
> go to safe 1
The safe 1 is open. In it, you see a cellphone 3, and a cellphone 2.
> put watch 1 in/on safe 1
You put the watch 1 in/on the safe 1.
```

## F   Detailed Evaluation Results on MiniWoB++

| Task Name | AdaPlanner | CC-Net [7] | WGE [11] | WebN-T5-3B [4] | RCI [8] |
|---|---|---|---|---|---|
| choose-list | 100 | 99 | 16 | 26 | 100 |
| click-button | 100 | 100 | 100 | 100 | 100 |
| click-button-sequence | 100 | 100 | 100 | 100 | 100 |
| click-checkboxes | 100 | 98 | 100 | 96 | 100 |
| click-checkboxes-large | 100 | 71 | 84 | 22 | 94 |
| click-checkboxes-soft | 80 | 95 | 94 | 54 | 72 |
| click-checkboxes-transfer | 98 | 99 | 64 | 63 | 100 |

| | | | | | |
|---|---|---|---|---|---|
| click-collapsible | 100 | 100 | 100 | 0 | 100 |
| click-collapsible-2 | 84 | 98 | 99 | 0 | 62 |
| click-color | 100 | 100 | 100 | 27 | 100 |
| click-dialog | 100 | 100 | 100 | 100 | 100 |
| click-dialog-2 | 100 | 100 | 100 | 24 | 100 |
| click-link | 98 | 99 | 100 | 100 | 100 |
| click-menu | 78 | 94 | n/a | 37 | 100 |
| click-option | 100 | 99 | 100 | 87 | 100 |
| click-scroll-list | 100 | 60 | n/a | 0 | 100 |
| click-shades | 100 | 100 | 99 | 0 | 100 |
| click-shape | 75 | 95 | 64 | 53 | 98 |
| click-tab | 100 | 100 | 100 | 74 | 100 |
| click-tab-2 | 85 | 98 | 98 | 18 | 74 |
| click-tab-2-hard | 78 | 98 | n/a | 12 | 76 |
| click-test | 100 | 100 | 100 | 100 | 100 |
| click-test-2 | 100 | 100 | 100 | 100 | 100 |
| click-widget | 100 | 100 | 93 | 100 | 98 |
| count-shape | 50 | 85 | 76 | 41 | 40 |
| email-inbox | 98 | 100 | 99 | 38 | 98 |
| email-inbox-forward-nl | 100 | 100 | n/a | 60 | 100 |
| email-inbox-forward-nl-turk | 100 | 100 | n/a | 33 | 94 |
| email-inbox-nl-turk | 90 | 100 | 93 | 23 | 98 |
| enter-date | 100 | 100 | 96 | 0 | 96 |
| enter-password | 98 | 100 | 100 | 97 | 100 |
| enter-text | 98 | 100 | 100 | 89 | 100 |
| enter-text-dynamic | 96 | 100 | 100 | 98 | 100 |
| enter-time | 96 | 97 | 90 | 0 | 100 |
| focus-text | 100 | 100 | 100 | 100 | 100 |
| focus-text-2 | 94 | 100 | 100 | 100 | 100 |
| grid-coordinate | 100 | 100 | 100 | 49 | 100 |
| guess-number | 88 | 100 | 0 | 0 | 20 |
| identify-shape | 96 | 100 | 100 | 88 | 76 |
| login-user | 100 | 100 | 100 | 82 | 100 |
| login-user-popup | 98 | 100 | n/a | 72 | 68 |
| multi-layouts | 84 | 100 | 100 | 83 | 72 |
| multi-orderings | 100 | 100 | 100 | 88 | 100 |
| navigate-tree | 82 | 99 | 99 | 91 | 86 |
| search-engine | 100 | 100 | 99 | 34 | 100 |
| simple-algebra | 82 | 75 | n/a | n/a | 100 |
| social-media | 82 | 90 | 100 | 21 | 98 |
| social-media-all | 100 | 75 | 1 | 0 | 100 |
| social-media-some | 90 | 85 | 42 | 2 | 90 |
| terminal | 98 | 0 | n/a | n/a | 100 |
| tic-tac-toe | 48 | 83 | 47 | 48 | 56 |
| use-autocomplete | 88 | 100 | 98 | 22 | 58 |
| use-spinner | 90 | 100 | 4 | 7 | 88 |

Table 8: Per-task success rate (%) of AdaPlanner, CC-Net [7], WGE [11], WebN-T5-3B [4], and RCI [8]. "n/a" signifies that the corresponding success rate is not reported in the original paper of the method. The nine tasks with feedback are marked in gray.

