# OpenReview forum: "AdaPlanner: Adaptive Planning from Feedback with Language Models"
_NeurIPS.cc/2023/Conference — NeurIPS 2023 poster_

### Official Review · Reviewer_aJpk · 2023-06-25

**Soundness:** 3 good
**Presentation:** 2 fair
**Contribution:** 3 good
**Rating:** 7
**Confidence:** 3

**Summary:**

LLMs have shown success as autonomous agents that make and execute plans in sequential decision problems. Existing methods either make open-loop plans, limiting adaptability to the environment, or closed-loop plans. Existing closed-loop methods, apart from DEPS, keep the plan static but simply modify immediate actions according to environment feedback, leading to potentially sub-optimal policies. The authors introduce AdaPlanner, a closed-loop LLM planner that additionally allows for *plan* refinement during the episode. The success of their method not only relies on this, but additionally code-style prompts and a skill-discovery mechanism for few-shot exemplars. AdaPlanner outperforms existing works while relying on far fewer demonstration examples from similar tasks.

**Strengths:**

- Empirically the authors show strong results with respect to sample efficiency and asymptotic performance.

- Many ablations make it easy to understand which components of the model lead to overall success.

- Conceptually simple approach.


**Weaknesses:**

- In the evaluation section, the baselines are glossed over. This makes it hard to comprehend the distinction between their approach and the baselines.
   - I’d recommend adding some of the Appendix descriptions to the evaluation section, and potentially referencing Table 1 more often.

- The authors use the term ‘hallucination’ a lot but do not define it.

- The authors discuss in- and out-of- plan refiners a lot before providing intuitive examples for when either would be necessary. Could the authors provide more examples earlier on in the paper?

- DEPS appears to be a relevant baseline. Could the authors include it or at least delve deeper into its limitations and why it is not appropriate?

- It appears that the largest contributor to the success of AdaPlanner, over existing approaches, is code style prompts and skill prompts. Wouldn’t it be worthwhile to apply those modifications to existing approaches, like Reflextion (Fig 4), and contrast?

- AdaPlanner prompts the LLM to correct any syntax errors. How important is this? Would be nice to include this ablation.


**Questions:**

- Line 80, could you define the output of pi, in the same way that you did for the planner?
- Line 81, shouldn’t it be P_t rather than P_{t - 1}?
- Lines 114 - 144 I think you’ve repeated the sentence twice.
- Line 216, what are the 6 task types?
- Line 132, how is N chosen and what’s its effect on performance?


**Limitations:**

AdaPlanner still requires demonstrations for learning. Would be worthwhile comparing with RL agents trained directly on the task, without any expert demonstrations.

---

> ### Author Rebuttal · Authors · 2023-08-10
>
> **[Weakness 1]**
>
> We appreciate the reviewer's comments. Due to the space limitation, we postponed the detailed introduction of the baseline to the Appendix 8.2. We evaluated AdaPlanner against a selection of representative baselines, both training-based and LLM-based.
> - In ALFWorld, we compared AdaPlanner to
>   - BUTLER: an imitation-learning baseline for ALFWorld[1];
>   - ReAct [9] and Reflexion [10]: two prominent LLM-based methods classified as *Implicit Closed-Loop Methods*.
> - In MiniWoB++, we evaluated AdaPlanner against
>   - CC-Net [11], WGE [12]: two training-based methods integrating supervised learning with reinforcement learning;
>   - WebN-T5-3B [13]:  a fine-tuned language model;
>   - RCI [2], a *Implicit Closed-Loop Method* for solving MiniWoB++ tasks.
>
> These methods reflect different perspectives (training-based and LLM-based) and offer a good reference frame for assessing how AdaPlanner enhances adaptive and sample-efficient planning.
>
> We will update our paper to show the distinction of these baseline and build a more detailed connection to Table 1.
>
>
> **[Weakness 2]**
>
> In the scope of LLM, “hallucination” commonly refers to *the produced content that is nonsensical or unfaithful to certain sources* [14,15,16,17]. When applied to the context of LLM for decision-making tasks, “hallucination” is specifically defined as two cases: 1) *the generated actions that are inadmissible in the environment*, or 2) *the unfaithful presumptions made by LLM about the environment settings*.
>
> An example of the first case is provided in *ReAct and Reflexion trajectory of Case 1*, Appendix 8.4.1, where the agent generates illegal actions; the one for the second case is in the *AdaPlanner Trajectory of Case 2*, Appendix 8.4.2, where the agent unfoundedly presumes the location of the watch.
>
> **[Weakness 3]**
>
> We appreciate the reviewer's comment on clarity. We will add an intuitive example from ALFWorld before the detailed introduction to these two refiners. We will also present Figure 2 earlier in the paper for better illustration.
>
> **[Weakness 4]**
>
> As shown in Table 1, DEPS relies on a training-based selector and requires additional data collection. It only utilizes past failures to refine its plans. In contrast, AdaPlanner can leverage both past successes and failures. Moreover, unlike AdaPlanner’s in-plan refinement, DEPS does not extract key information from observations, which may make it less adaptive than AdaPlanner.
>
> DEPS was constructed around the OpenAI Codex API, which has been deprecated. DEPS is also primarily designed for Minecraft and poses difficulties for transfer to ALFWorld and MiniWoB++.
>
> **[Weakness 5]**
>
> ReAct and Reflexion are notable for their use of natural language-based open-loop/in-plan reasoning and planning. The code interface and skill discovery are two of our major contributions over these methods. Although these two techniques are generally compatible with the existing methods, once these features were incorporated into the existing approaches, the modified variants would convert to new methods towards our method, which no longer accurately reflect the performance of the original methods as described in the papers. This would diminish the significance and novelty of the proposed AdaPlanner with our own techniques.
>
> **[Weakness 6]**
>
> We added an ablation study on this component as follows:
> | Environment                      | Baseline | w/o Code Check |
> |-----------------------------------|---------------|----------------|
> | ALFWorld            | 80.60          | 79.85          |
> | MiniWoB++ | 92.87         | 91.92          |
>
> The table above shows the success rate (%) ablating the code check. As shown, the code check contributes to enhancing the overall performance. This improvement is relatively marginal, because of the capability of the LLMs (GPT-3 and GPT-3.5) to generate code almost free of syntax errors. Upon further investigation, we found that only 1.49% of the code generated for ALFWorld contained syntax errors, thus making the code check not the primary contribution to the performance improvement. Instead, as detailed in Figure 4, the main contributions of our study are the proposed closed-loop structure, code-styled prompting, and skill discovery. These mechanisms collaboratively foster adaptive and sample-efficient decision-making.
>
> **[Question 1]**
>
> $\pi(\cdot|g, c_t, P_t)$ generates the action at $t+1$ step conditioned on a given plan  $P_t$.
>
> **[Question 2]**
>
> It should be $P_{t-1}$ as we were discussing the $t$-th step.
>
> **[Question 3]**
>
> Thank you for this comment. We will revise the sentence in our updated paper.
>
> **[Question 4]**
>
> The 6 task types are ```Pick, Clean, Heat, Cool, Examine, Pick two```. We provided details about these tasks in Appendix 8.1 (line 407-415).
>
> **[Question 5]**
>
> The LLM agent automatically determines the value of $N$ during plan generation. It's not a manually set parameter but rather task-specific.
>
>
> **[Limitations]**
>
> The skill discovery mechanism can greatly alleviate the need for demonstration. In the ablation studies (Figure 4d), we applied a zero-shot prompting for MiniWoB++ by *omitting any demonstrations*. AdaPlanner then successfully finds feasible solutions over 21 tasks and enhances the overall success rate by 15%. As indicated in Figure 3, Incorporating skill discovery can effectively reduce the number of demonstrations required to achieve satisfactory performance.
>
> Generally, RL agents using expert demonstrations outperform those without them. In our evaluation, we compared AdaPlanner with several RL-refined imitators, such as BUTLER, CC-Net, and WGE, which employ 100k, 23k, and 10 demonstrations per task, respectively. Comparing AdaPlanner with these baselines underscores AdaPlanner's superior performance and suggests its potential superiority over baselines that do not use demonstrations.
>
> Our future work will focus on enhancing AdaPlanner to perform well even without demonstrations.

---

> > ### Comment · Reviewer_aJpk · 2023-08-14
> > **Response to Rebuttal**
> >
> > I thank the authors for their detailed rebuttal. I appreciate the additional ablation and the comparisons made with methods that do not leverage demonstrations. In line with my review, I believe this is a good paper and will keep my score as is.

---

> > > ### Author Response · Authors · 2023-08-19
> > >
> > > Thanks for reading our rebuttal and providing valuable feedback. We will incorporate these additional results and discussions into the updated version of our paper.

---

### Official Review · Reviewer_oy34 · 2023-06-29

**Soundness:** 2 fair
**Presentation:** 2 fair
**Contribution:** 2 fair
**Rating:** 6
**Confidence:** 4

**Summary:**

Briefly summarize the paper and its contributions. This is not the place to critique the paper; the authors should generally agree with a well-written summary.

The paper proposes AdaPlanner, an LLM-based adaptive planner for text-based sequential decision-making tasks. The planner is adaptive in the sense that it can refine the generated plan/policy based on feedback.

The contributions made in this paper include the following
1. interacting with the environment with LLM in the loop
2. a code-style prompt is engineered for LLMs to output a policy
3. refining the LLM policy for the current task based on feedback
4. prompt tuning for new tasks based on previous interaction (termed skill discovery)

The proposed AdaPlanner is evaluated on two text-based sequential decision-making environments ALFWorld and MiniWoB++. Their experiments indicate that with feedback, LLMs can adapt the plan.


**Strengths:**


* The paper is well written.
* The paper focuses on extremely relevant and signifcant problems.


**Weaknesses:**

* I find the paper lacks significant details. Please see the next section for the list of questions.
* The paper employs sloppy mathematical notations.
* The paper lacks the rigor of scientific evaluation.
* Paper misses all references to LLM-based approaches for planning with PDDL. The one that I find most relevant for code generation is "Generalized Planning in PDDL Domains with Pretrained Large Language Models, Tom Silver, Soham Dan, Kavitha Srinivas, Joshua B. Tenenbaum, Leslie Pack Kaelbling, Michael Katz”


**Questions:**


**Major**

1. How is the programmatic response from LLM converted to action responses? Did the conversion require manual intervention? For instance, Figure 2 has an indentation error which would result in a wrong plan. Were such indentation errors evaluated manually? Can authors provide a list of errors made by LLMs?
1. In line 167, what does an alignment between ‘anticipated plan’ and environment mean? How does the AdaPlanner observe the alignment?
1. Can authors provide details about the size of the task used in the prompt (for samples) vs the size of the task that was successfully solved by AdaPlanner? To establish the claim of sample efficiency, it is important to understand if the planner is able to efficiently plan for tasks that are significantly different from the prompts.
1. The X-axis in Figure 3 indicates `# Samples per task`. Is this the number of samples provided for each trajectory? Or sum?
1. What was the length of plans or length of trajectories generated by AdaPlanner vs other approaches? To claim the effectiveness of the AdaPlanner, it is important to compare the length of successful trajectories.
1. For skill discovery, how is the solution converted to the skill? How are skills represented? How large is the skill memory?  Were the discovered skills included in the count of samples used for training as they are training samples for the next set of trajectories?
1. It is not clear how skills are filtered and what criteria are used for the evaluation and ranking of skills.
1. What is the connection between skill discovery and prompt tuning?
1. The success rate of "With SD" in Figure 4d looks significantly reduced from  Figure 4a. Were different settings used for theses experiments?
1. At various places, the paper mentions "environment feedback". In my opinion, this is a misnomer. The feedback is not from the environment. The environment just provides the next observation, the feedback is generated by the agent itself. And the use of observation to refine a plan or next action is quite standard practice in RL. I would highly recommend dropping the term feedback from the title.
1. The use of term plan and policy is a little confusing. A plan is a sequence of actions. A policy is a mapping from states to actions. By this definition, the `solution()` function is as a policy. In preliminaries, the planning policy ($\rho$) is conditioned on a previous plan $P_t$. However, the appendix describes the refinement prompt using the assertion error (instead of `solution()`). Isn't the assertion error providing information about the policy (the `solution()` function)? So I am confused by the terminologies. Is the $\rho$ refined conditioned on the policy or the plan? The usage of these terms is also confusing in the Preliminary section. Request authors to precisely define the mathematical notations and highlight what they represent in the examples.

**Minor**

12. In line 387, there are extra curly braces.
12. The notation $\rho$ is used in line 73 but introduced much later.
12. As the context $c_t$ is defined as a sequence of action and observations from time step $0$ to $t$, it is not clear what $c_{>t}$ means (in line 116).
12. Open-Loop system in Figure 1 should have an arrow going from env to planner with $o_1$.
12. Statement in Line 144 "To generate a plan .." looks like a repetition of Line 141 "To generate an initial plan..."
12. In line 116, if $h_t$ is obtained from $c_t$ then would it not be captured in $c_{>t}$? An example of $h_t$ would help better understand the proposed update.
12. In line 73, as $\rho$ is defined using $\Delta(A^{T})$. But the length $T$ is not fixed.
12. In line 73 $\rho$ is defined where a plan is conditioned only on observation and goal. However, later it is conditioned on the context, plan, and goal.





**Limitations:**


* The evaluations are restricted to text-based sequential decision-making problems and task where the inadmissible actions do not cause drastic changes in the environment. On the contrary, inadmissible actions are like no-ops. Further, the paper does not present analysis of plan length. Hence, the analysis is limited to zero risk environments.
* The claim made in the abstract about skill discovery mechanism enabling agent to plan with fewer task demonstration is not substantiated in the evaluations. Evaluation in Fig. 4d only established improvement in success rate, not sample efficiency.

---

> ### Author Rebuttal · Authors · 2023-08-10
>
> Due to the 6000 characters limit, we only address the important questions. For detailed clarification, we will provide them during the discussion period.
>
> **[Weakness 1]** Due to the page limit, we primarily discussed the main components of AdaPlanner: closed-loop structure, code-styled prompting, and skill discovery in the main body. We have included additional details in the supplementary section to provide a more comprehensive understanding.
>
> In response to the reviewer's list of questions regarding details and notations, we will address each of them separately in the following section. For some of the questions, we can incorporate clarifications in the updated version of our paper.
>
> **[Weakness 2]** In Section 2, we have provided a thorough explanation of the mathematical notations for understanding our method. We have defined planning policies for both open- and closed-loop systems in detail.
>
> For the comment "sloppy mathematical notations," we would appreciate it if the reviewer could provide specific examples. This will allow us to address the concern accurately and make any necessary revisions.
>
> **[Weakness 3]** We strongly disagree with the statement that "the paper lacks the rigor of scientific evaluation". Our evaluation is rigorous in the following aspects:
> - Benchmarks: We provided thorough evaluations spanning two representative and widely-accepted decision-making environments;
> - Baselines: We've contrasted AdaPlanner with prevailing methods and showed our approach outperforms existing methods in success rate and sample efficiency.
> - Ablation Studies: We validated contribution of each component within AdaPlanner.
> - Reproducibility: We've included extensive technical details to ensure strong reproducibility.
>
> We firmly believe our research upholds rigorous scientific evaluation standards.
>
> **[Weakness 4]** Most of the existing work along this line of research, including [4, 5, 6] is still open-loop with PDDL. The only exception is [3], as the reviewer suggested. But it was impossible for us to cite in our submission because the paper was uploaded to ArXiv on May 18, 2023 --- after the submission deadline.
>
> Meanwhile, there are significant differences between [3] and our work. In [3], the plan formulated on the training set lacks task-specific refinement during the execution of the evaluation task. In contrast, our AdaPlanner dynamically refines the plan and adapts to various feedback.
>
> Our mechanisms in AdaPlanner could be applied to PDDL, but this extension is out of the current paper's scope and could be explored in the future.
>
> **[Q1.1]** As described in line 161-164, we formulated an environment interface that 1) grounds the generated actions from the solution() function to the admissible actions in the environment, and 2) routes the observation from the environment back to the code as a return value.
>
> **[Q1.2]** No. The interface described above automatically carries out this conversion without manual intervention.
>
> **[Q1.3]** Thanks for pointing out the indentation problem in Figure 2. We found that it was a typo made by the authors when making this figure, not the LLM. We will correct this typo in the updated version.
>
>
> **[Q2]** The alignment means the execution of the plan proceeds as expected and no assertion is triggered. This is consistent with our definition of in-plan feedback introduced in line 44-46. Note that there is no in-plan alignment, in-plan refinement is only used for extracting key information from feedback.
>
> **[Q3]** We added evaluation results in MiniWoB++ as follows:
>
> | Task group                                      | With Feedback  | No Feedback  | All |
> |-------------------------------------------------|----------------------------------|---------------------------------|---------------------------|
> | # Tasks solved  | 410.00                           | 2,050.84                        | 2,460.84                  |
> | # Samples in the prompt           | 13                               | 25                              | 38                        |
> | # Sample / # solved tasks (%)             | 3.17                             | 1.22                            | 1.54                      |
>
> In MiniWoB++, the ratio # Sample / # solved tasks decreases to approximately 3%. This discussion and Figure 3 jointly show AdaPlanner’s sample efficiency.
>
> **[Q4]** ``# Samples per task`` refers to the number of expert demonstrations provided for each task type.
>
> **[Q5]** The comparison of the average trajectory lengths in ALFWorld is presented as follows:
> |   Method   |  Trajectory length (step) |
> |:----------:|:---------------:|
> |    ReAct   |    25.81      |
> |  Reflexion |     18.90      |
> | AdaPlanner |   15.60      |
>
> AdaPlanner generally requires fewer steps to complete the tasks compared to ReAct and Reflexion.
>
> **[Limitation 1]** We conducted an analysis of the trajectory length, detailed in our response to *Question 5*, which underscores the effectiveness of AdaPlanner.
>
> The generalization to non-zero-risk environments for LLM agents is an interesting future direction. However, this is beyond the scope of our paper, and there is no widely-accepted benchmark for these tasks yet.
>
> While we have focused on text-based environments, our method is not limited to them. For example, integrating vision-language models like CLIP could potentially allow AdaPlanner to interact with more visually complex environments.
>
> **[Limitation 2]** We added ablation studies in MiniWoB++. The method with skill discovery requires only 15 samples to outperform the variant without skill discovery, even though the latter used twice as many samples. It is evident that skill discovery enhances sample efficiency.
> | \# samples                  | 38    | 30    | 20    | 15    | 0     |
> | --------------------------- | ----- | ----- | ----- | ----- | ----- |
> | With SD (%)    | 92.87 | 84.06 | 79.17 | 75.17 | 60.38 |
> | Without SD (%) | 82.40 | 73.58 | 68.70 | 64.70 | 45.47 |

---

> > ### Author Response · Authors · 2023-08-12
> >
> > **Please check the remaining part of our rebuttal to Reviewer oy34 as follows:**
> >
> > **[Q1.4]** We summarized the errors that have occurred through the evaluation in ALFWorld as follows:
> > | Type              | Description                                                              |
> > | ----------------- | ---------------------------------------------------------------------------------------------------------------------------------------------------------------------------------------------------------------------------------------------------------------------------------------------------------------------------------------------------- |
> > | Indentation error | Occasionally, the code may contain indentation errors. This error can be addressed by the Code Check. |
> > | Reference error   | The ```start_from``` may be misconfigured and the last breakpoint are not properly loaded. This error can be mitigated by out-of-plan refinement.                                                              |
> > | Incomplete code   | Due to the token limit, generated code might sometimes be truncated. We can adopt newer models that support extended context lengths, such as gpt-3.5-turbo-16k.                                                         |
> >
> > **[Q6.1/6.2]** We use each solution as a demonstration in the prompt to solve similar tasks. If including the solution boosts the success rate, we will keep the solution in the skill memory and add it to the prompt for future task-solving.
> >
> > **[Q6.3]** The skill memory can potentially be large, depending on the task complexity. The automatic adjustment of the skill memory's size is an important feature that we will investigate in the future.
> >
> > **[Q6.4]** AdaPlanner is purely prompting-based and there is no training process involved. We add the discovered skills to the prompts. The skills are actively discovered by the agent, thus not included in the count of expert demonstrations.
> >
> > **[Q7]** As illustrated in line 209-213, we assess the effect of using each discovered skill as the demonstration in the prompt to solve similar tasks. If adding a candidate solution boosts the success rate on these tasks, it's added to the memory of discovered skills; if not, it's discarded.
> >
> > **[Q8]** The successful skills discovered by AdaPlanner will be added to the prompt. It could be understood as a method for prompt tuning. However, we emphasize that this procedure is automatically completed by LLM itself during the planning.
> >
> > **[Q9]** Yes. The detailed settings for these two figures were mentioned in line 227-229, 442-445. In Figure 4a, we adopt the setting of prompted samples as in Table 4. In Figure 4d, we only provide one sample of the simplest task (pick) and use skill discovery to explore skills for the rest five tasks. The difference in the success rate of “with SD” in Figure 4d originates from this setting difference.
> >
> > **[Q10]** We use “environment feedback” to indicate any outcome provided by the environment, including the observations.
> >
> > **[Q11]** The assertion error provides information about the plan rather than the policy. For example, in ALFWorld, the error message reports which action within the plan has been executed and the error occurs (e.g., ```Error in [Step X], …```). This information corresponds to $P_{t-1}$ as in the definition of $\rho(P_t|g, c_t, P_{t-1})$ in line 119.
> >
> > **[Typos] Q12, 13, 14, 15, 16, 19** Thank you for the comments. We will revise these typos and notations in the updated version of our paper.
> >
> > **[Q17]** The $h_t$ would be included in $c_{>t}$. For example, the agent identifies the target object ```book 1``` from the environment observation ```On drawer 2, you see a book 2, and a keychain 1.``` Here $h_t$ is the identifier information which is then used for future actions.
> >
> > **[Q18]** We fix $T$ as the step limit of the environments. For example, in ALFWorld, this number is set to 50.

---

> > > ### Comment · Reviewer_oy34 · 2023-08-16
> > >
> > > Thank you for your thorough response.  I still find that literature that uses LLM with PDDL is relevant to this work and should be cited. This is a fast-moving field and hence keeping track of all work & its chronology is clearly impossible, but **not** citing **any** of the work done in that field is un-scholarly.  I appreciate the addition of the empirical results that study sample efficiency. I believe the response to Limitation 2 (and other additional results) addresses Weakness 3. I will update my score accordingly.

---

> > > > ### Author Response · Authors · 2023-08-19
> > > >
> > > > Thank you for taking the time to review our rebuttal and offering insightful feedback. We will update our paper with the additional results and discussions and add the literature review on methods leveraging LLM with PDDL.

---

### Official Review · Reviewer_GDYQ · 2023-07-04

**Soundness:** 3 good
**Presentation:** 3 good
**Contribution:** 2 fair
**Rating:** 5
**Confidence:** 4

**Summary:**

The paper presents AdaPlanner, a closed-loop planning method that uses a large language model (LLM) to solve tasks in text-based environments. AdaPlanner operates by decomposing a complex task into manageable sub-goals and predicting environmental feedback for each. During execution, it refines its actions based on the feedback received from the environment. AdaPlanner operates solely via prompting, eliminating the need for a dedicated training phase and reducing its computational cost. The paper demonstrates that AdaPlanner consistently outperforms existing baselines, achieving state-of-the-art performance in ALFWorld tasks and MiniWoB++ tasks.

**Strengths:**

- AdaPlanner introduces a novel approach to task-solving in text-based environments using a large language model. It stands out for its closed-loop planning method and its ability to decompose tasks into manageable sub-goals.
- The paper is well-written and clear. The authors have done a good job of explaining complex concepts and methodologies in an understandable manner.
- The work presents a new way of leveraging large language models for task-solving in text-based environments. The results show that AdaPlanner can effectively leverage feedback to refine its plans and enhance its performance.

**Weaknesses:**

- The part about skill discovery is not described very clearly, and I still cannot understand the details of the skill discovery module well.
- The author compared the version without a code interface in the experiment, but it seems that they did not specifically show the prompt after removing the code interface. At the same time, as an ablation experiment, it is also necessary to analyze the effects of specific components in the code interface.
- The phenomenon that GPT-3 performs better than GPT-3.5 is interesting, but it seems that the paper only compares GPT-3 and GPT-3.5 in Alfworld, without conducting the same experiments in MiniWoB++ to further support the conclusion. And the author's hypotheses about this phenomenon (the smaller scale of GPT3.5) lacks specific analysis or literature references to support it.

**Questions:**

- In the experiment, what is the proportion of in-plan and out-of-plan occurrences? How will this proportion change over time? This should be a necessary indicator for understanding the two refiners.
- On MiniWoB++, will there be better performance from GPT-3 than GPT-3.5?
- Is there still a necessity for AdaPlanner in larger-scale LLMs, such as models like GPT4 with better self-refining capabilities?

**Limitations:**

- As mentioned above, this paper still needs more experiments and analysis to further validate the rationality of its methods, as well as the observed phenomena and corresponding hypotheses.

---

> ### Author Rebuttal · Authors · 2023-08-10
>
> **[Weakness 1]**
>
> Thank you for your valuable feedback. In the skill acquisition stage, AdaPlanner harnesses adaptive closed-loop planning to solve unseen tasks using limited or no demonstrations. The successful solutions found in this trial-and-error process are called the candidate discovered skills and are gathered in a pool.
>
> During the following skill filtering stage, we assess the effect of using each discovered skill from the pool as the demonstration in the prompt to solve similar tasks. If a candidate solution boosts the success rate, it is added to the memory of discovered skills; if not, it is discarded.
>
> **[Weakness 2]**
>
> For the *prompts without code interface (CI)*:
> The initial planning phase is prompted as follows:
>
> ```
> Play as a planner. Make a general plan to finish the task by referring to the paradigm that solved a similar task.
>
> Here is an example:
>
> <example>
>
> You need to generate a new plan by transferring the given paradigm to the new task.
>
> [GIVEN PARADIGM]
>
> <paradigm>
>
> [NEW TASK]
>
> <task>
>
> ```
>
> The <example> contains human demonstrations of general plans as follows. Note that the question marks like (id?) indicate the information that will be completed through feedback.
>
> ```
> Your task is to: heat some egg and put it in diningtable.
>
> 1. Search_and_find egg (id?).
> 2. Take egg (id?) from the place it was found (name? id?).
> 3. Go_and_heat egg (id?) to microwave 1 to heat the egg (id?).
> 4. Go_and_put egg (id?) in/on the diningtable 1.
> ```
>
> For *Ablation study on code interface components*:
>
> We conducted the ablation studies on several components of the code interface, such as the assertion that allows for the out-of-plan refinement and the breakpoint start_from that enables the refine-then-resume mechanism.
>
> | Environment | Baseline | w/o Code Check | w/o assertion|
> |:----------------:|:-----------:|:--------------------:|:-------------:|
> |ALFWorld |80.60|79.85|75.12|
> |MiniWoB++ |91.11|89.78|77.78|
>
> The table above shows the success rate (%) ablating ablating assertion and the breakpoint ```start_from```. In the table, removing any component results in a drop in performance, which signifies the effectiveness of these components in AdaPlanner's design. Notably, when the code interface is fully substituted with natural language, as previously detailed, there is a significant reduction in the performance across both environments. This significant performance drop underscores the essential role of the code interface in AdaPlanner.
>
> **[Weakness 3]**
>
> We evaluated AdaPlanner and RCI on gpt-3.5-turbo and compared the results with those obtained on text-davinci-003:
>
> | Method                       | With Feedback (9 tasks) | No Feedback (44 tasks) | All (53 tasks) |
> |-------------------------------|-------------------------|------------------------|----------------|
> | RCI (gpt-3.5-turbo)           | 70.89                   | 76.36                  | 75.43          |
> | RCI (text-davinci-003)        | 81.56                   | 92.68                  | 91.00             |
> | AdaPlanner (gpt-3.5-turbo)    | 75.56                   | 78.05                  | 77.63          |
> | AdaPlanner (text-davinci-003) | 91.11                   | 93.22                  | 92.87          |
>
> The table above shows the success rate (%)  on two subsets of tasks in MiniWoB++. We observe a noticeable performance drop of both methods on gpt-3.5-turbo, which coincides with the results obtained on ALFWorld (Table 2).
>
> We also found a similar hypothesis drawn based on various experiments in [7]. Given that OpenAI does not publicly disclose the technical details or source code for its GPT models, a definitive validation of our hypothesis remains challenging. As indicated in OpenAI documentation, the gpt-3.5-turbo model is primarily optimized for human conversation tasks, which could potentially compromise its performance on tasks such as code generation and reasoning [8].
>
>
> **[Question 1]**
>
> We investigated the occurrence (average times per task) of both in-plan and out-of-plan refinements with our analysis as shown below:
> | Environment                      | In-plan refinement | Out-of-plan refinement | Proportion (in/out) |
> |-----------------------------------|--------------------|-------------|---------------------|
> | ALFWorld         					| 2.83               | 6.40        | 0.44                |
> | MiniWoB++ 					 | 0.33               | 1.78        | 0.19                |
>
> The table illustrates that out-of-plan refinement is invoked more frequently in both environments, emphasizing its important role in AdaPlanner's planning process. The average occurrence for both types of refinements varies between ALFWorld and MiniWoB++. This can be attributed to the differences in the average lengths of the trajectories in ALFWorld and MiniWoB++, which stand at 15.60 and 5.40 steps, respectively.
>
> The proportion of in-plan to out-of-plan refinements is task-specific. In tasks with extensive environmental observations in natural language, in-plan refinement might be more prevalent. On the other hand, analyzing this proportion 'over time' within a task may be challenging because neither type of refinement occurs densely enough to make such a temporal analysis meaningful.
>
> We will include these findings and discussions in the updated version.
>
>
> **[Question 2]**
>
> Yes. Please refer to our response above on *Weakness 3*.
>
> **[Question 3]**
>
> We would like to emphasize that the proposed closed-loop structure and adaptive refinement in AdaPlanner are fundamentally compatible with a wide range of LLMs. AdaPlanner provides such a robust closed-loop planning framework and is supposed to assist larger-scale LLMs in further enhancing their performance.
>
> However, we haven't been granted access to the GPT-4 API at this time. Once it becomes available to us, we will carry out additional evaluations of AdaPlanner on GPT-4 and will update our paper to reflect any significant findings or discussions that arise from this analysis.

---

> > ### Comment · Reviewer_GDYQ · 2023-08-14
> >
> > Thanks for the author's response. Most of my concerns have been addressed, for now I will maintain my current score and continue to pay attention to other reviews and discussions.

---

> > > ### Author Response · Authors · 2023-08-19
> > >
> > > Thank you for offering feedback on our rebuttal. We will update our paper and include the additional results and discussions.

---

### Official Review · Reviewer_4ZaS · 2023-07-07

**Soundness:** 3 good
**Presentation:** 3 good
**Contribution:** 3 good
**Rating:** 6
**Confidence:** 3

**Summary:**

This paper looks at explicit closed-loop systems with LLMs for adaptive planning utilizing environmental feedback. They showcase better planning performance on ALFWorld and MiniWOB++ environments over existing state-of-the-art works like ReAct and Reflexion.

**Strengths:**

The paper is well written and the experiments are thorough. They present an interesting improvement over the current works like ReAct and Reflexion.


**Weaknesses:**

1. The kind of tasks in these domains don’t seem to have interaction resolution where there are multiple conflicting causal links from the initial to the goal state which have to be resolved (including negative interactions between subgoals). This could also lead to the human demonstrations helping significantly with the  It would be useful to analyze the performance of AdaPlanner specifically in such cases.

2. I think non-ergodic environments could clearly pose danger to such agents. It would be interesting to see how AdaPlanner can perform against ReAct or Reflexion in such environments.


**Questions:**

1. Given that the LLM seems to verify the plan to determine its feasibility, what was its efficiency in those assessments? Are there any results pertaining to that?

2. Is there any classification of the tasks with respect to their hardness?

3. For how many of these tasks did the human expert demonstration solve the task?


**Limitations:**

The authors have addressed some of the limitations. I have provided some limitations in the weaknesses section.

---

> ### Author Rebuttal · Authors · 2023-08-10
>
> **[Weakness 1]**
>
> Thank you for your insightful comments. We assume “conflicting causal link” is referring to the existence of the non-revocable actions and states.” In MiniWoB++, a significant number of tasks indeed present multiple conflicting causal links. For example, a mistaken click in the task ```login-user-popup``` can directly lead to task failure.  As shown in our evaluation (Appendix 8.5), when provided with a single human demonstration for this task, AdaPlanner achieves a success rate of 98%, which is comparable to the state-of-the-art training-based baselines and outperforms all LLM baselines. We can update our paper to include specific case studies that elaborate on such cases.
>
> **[Weakness 2]**
>
> We assume the “non-ergodic environments” refer to environments that present irrevocable actions. Within this context, both ALFWorld and MiniWoB++ are non-ergodic. In ALFWorld, several actions, such as ```clean```, ```cool```, and ```heat``` will pose an irreversible effect on the object. Also, most MiniWoB++ tasks encompass non-ergodic properties as several actions, such as clicking a button, are irreversible. Within this context, AdaPlanner has shown good overall performance in comparison with ReAct, Reflexion, and RCI. Our future work will focus on evaluating these approaches in other representative non-ergodic environments.
>
>
> **[Question 1]**
>
> In AdaPlanner, the proposed code check and out-of-plan refinement determine and improve the feasibility of the generated plan. The following table shows the success rate (%) of AdaPlanner in ALFWorld and MiniWoB++ (in 9 tasks with feedback), ablating the out-of-plan refinement (assertion component) and code check.
>
> | Environments | Baselines | w/o Code Check | w/o assertion|
> |:----------------:|:-----------:|:--------------------:|:---------:|
> |ALFWorld (134 tasks)|80.60|79.85|75.12|
> |MiniWoB++ (9 tasks)|91.11|89.78|77.78|
>
> As shown in the table, both the code check and out-of-plan refinement contribute to enhancing the success rate, with the latter demonstrating a particularly significant impact on AdaPlanner’s overall performance. This result shows the importance of the proposed assertion mechanism in efficiently validating the current plan's feasibility, thereby facilitating adaptive and robust planning through continuous feedback and refinement.
>
> **[Question 2]**
>
> In ALFWorld, the tasks are categorized based on the length of the solution trajectories.[1]:
> | Hardness                                                  | Easy                          | Medium                                  | Medium                                 | Medium                                 | Hard                         | Hard                                                |
> | --------------------------------------------------------- | ----------------------------- | --------------------------------------- | -------------------------------------- | -------------------------------------- | ---------------------------- | --------------------------------------------------- |
> | Task                                                      | Pick                          | Clean                                   | Heat                                   | Cool                                   | Examine                      | Pick two                                            |
> | general solution chain                                    | find-> pick -> goto -> put | find->pick->goto->clean->goto->put | find->pick->goto->heat->goto->put | find->pick->goto->cool->goto->put | find->pick->find->examine | find->pick->goto->put->find->pick->goto->put |
> | Average length of AdaPlanner solution trajectories (step) | 10.79                         | 13.45                                   | 17.61                                  | 13.33                                  | 21                           | 20.71                                               |
>
> In MiniWoB++, we follow the difficulty classification outlined in [2], arranging the 53 tasks into three tiers (easy, medium, and hard) based on the range of success rates. This categorization is illustrated in the following table:
> | Hardness           |     Easy     |    medium   |     hard    |
> |--------------------|:------------:|:-----------:|:-----------:|
> | Success rate range |   [1, 0.9]   |  (0.9, 0.6] |   (0.6, 0]  |
> | Example task       | click-widget | click-tab-2 | count-shape |
>
> We will update our paper with this specification of hardness classification in the evaluation studies.
>
> **[Question 3]**
>
> In ALFWorld, each of the six task types is provided with a single human expert demonstration, resulting in a total of 6 expert demonstrations across all 134 tasks. In MiniWoB++, we use 38 human demonstrations spread over 53 tasks. All LLM-based approaches we tested, including Adaplanner, employ different task settings when creating these expert demonstrations, which precludes their direct application to solving tasks in the test set. For instance, an expert demonstration in ALFWorld might involve receptacles and objects that do not align with those in the actual test cases.

---

> > ### Comment · Reviewer_4ZaS · 2023-08-21
> >
> > Thank you for the detailed response. Most of my concerns have been addressed.

---

### Author Rebuttal · Authors · 2023-08-10

**References**

[1] M. Shridhar, X. Yuan, M.-A. Cote, Y. Bisk, A. Trischler, and M. Hausknecht. ALFWorld: Aligning text and embodied environments for interactive learning. In International Conference on Learning Representations, 2021.

[2] G. Kim, P. Baldi, and S. McAleer. Language models can solve computer tasks. arXiv, page
2303.17491v1, 2023.

[3] Generalized Planning in PDDL Domains with Pretrained Large Language Models, Tom Silver, Soham Dan, Kavitha Srinivas, Joshua B. Tenenbaum, Leslie Pack Kaelbling, Michael Katz

[4] Liu, B., Jiang, Y., Zhang, X., et al. 2023. LLM+P: Empowering Large Language Models with Optimal Planning Proficiency. arXiv.

[5] Tom Silver, Varun Hariprasad, Reece S Shuttleworth, Nishanth Kumar, Tomás Lozano-Pérez, and Leslie Pack Kaelbling. Pddl planning with pretrained large language models. In NeurIPS 2022 Foundation Models for Decision Making Workshop, 2022.

[6] Vishal Pallagani, Bharath Muppasani, Keerthiram Murugesan, Francesca Rossi, Lior Horesh, Biplav Srivastava, Francesco Fabiano, and Andrea Loreggia. Plansformer: Generating symbolic plans using transformers. arXiv preprint arXiv:2212.08681, 2022.

[7] Ye, Junjie, et al. "A comprehensive capability analysis of gpt-3 and gpt-3.5 series models." arXiv preprint arXiv:2303.10420 (2023).

[8] OpenAI. OpenAI Documentation: GPT-3.5 models. Retrieved May 16, 2023, from https://platform.openai.com/docs/models/gpt-3-5

[9] S. Yao, J. Zhao, D. Yu, N. Du, I. Shafran, K. R. Narasimhan, and Y. Cao. React: Synergizing reasoning and acting in language models. In The Eleventh International Conference on Learning Representations, 2023.

[10] N. Shinn, B. Labash, and A. Gopinath. Reflexion: an autonomous agent with dynamic memory and self-reflection. arXiv preprint arXiv:2303.11366, 2023.

[11] P. C. Humphreys, D. Raposo, T. Pohlen, G. Thornton, R. Chhaparia, A. Muldal, J. Abramson, P. Georgiev, A. Goldin, A. Santoro, and T. Lillicrap. A data-driven approach for learning to control computers. arXivProceedings of the 39th International Conference on Machine Learning, Baltimore, Maryland, USA, PMLR 162, 2022, page 2202.08137v2, 2022.

[12] E. Z. Liu, K. Guu, P. Pasupat, and P. Liang. Reinforcement learning on web interfaces using
workflow-guided exploration. In International Conference on Learning Representations, 2018.

[13] I. Gur, O. Nachum, Y. Miao, M. Safdari, A. Huang, A. Chowdhery, S. Narang, N. Fiedel, and A. Faust. Understanding html with large language models. arXiv, page 2210.03945v1, 2022.

[14] Katja Filippova. 2020. Controlled Hallucinations: Learning to Generate Faithfully from Noisy Data. In Proceedings of the 2020 Conference on Empirical Methods in Natural Language Processing: Findings. 864–870.

[15] Joshua Maynez, Shashi Narayan, Bernd Bohnet, and Ryan McDonald. 2020. On Faithfulness and Factuality in Abstractive Summarization. In Proceedings of the 58th Annual Meeting of the Association for Computational Linguistics. 1906–1919.

[16] Ankur Parikh, Xuezhi Wang, Sebastian Gehrmann, Manaal Faruqui, Bhuwan Dhingra, Diyi Yang, and Dipanjan Das. 2020. ToTTo: A Controlled Table-To-Text Generation Dataset. In Proceedings of the 2020 Conference on Empirical Methods in Natural Language Processing (EMNLP). 1173–1186.

[17] OpenAI. 2023. GPT-4 Technical Report. arXiv preprint arXiv: 2303.08774.

---

### Decision · Program_Chairs · 2023-09-21

**Decision:**

Accept (poster)

**Comment:**

The authors introduce a closed loop planner for textbased environments (miniwob and alfworld) with state of the art results.  The underlying model uses code based prompting.  Crucially the process is both iterative in that it's closed loop and that it performs refinement/corrections of its own plans. There is also a skill acquisition process by which a limited number of human demonstrations initialize the model to explore the action space and determine whether inclusion of discovered paths into the prompt improves success. The proposed approach outperforms previous algorithms that leverage the same underlying model.